# Membrane protein MHZ3 regulates the on-off switch of ethylene signaling in rice

Xin-Kai Li [1,2], Yi-Hua Huang [1], Rui Zhao[1], Wu-Qiang Cao[1], Long Lu[3], Jia-Qi Han [1], Yang Zhou [1], Xun Zhang[1,2], Wen-Ai Wu[1,2], Jian-Jun Tao[1], Wei Wei[1], Wan-Ke Zhang [1], Shou-Yi Chen [1], Biao Ma[4], He Zhao[5] ✉, Cui-Cui Yin [1] ✉ & Jin-Song Zhang [1,2] ✉

Ethylene regulates plant growth, development, and stress adaptation. However, the early signaling events following ethylene perception, particularly in the regulation of ethylene receptor/CTRs (CONSTITUTIVE TRIPLE RESPONSE) complex, remains less understood. Here, utilizing the rapid phospho-shift of rice OsCTR2 in response to ethylene as a sensitive readout for signal activation, we revealed that MHZ3, previously identified as a stabilizer of ETHYLENE INSENSITIVE 2 (OsEIN2), is crucial for maintaining OsCTR2 phosphorylation. Genetically, both functional MHZ3 and ethylene receptors prove essential for OsCTR2 phosphorylation. MHZ3 physically interacts with both subfamily I and II ethylene receptors, e.g., OsERS2 and OsETR2 respectively, stabilizing their association with OsCTR2 and thereby maintaining OsCTR2 activity. Ethylene treatment disrupts the interactions within the protein complex MHZ3/receptors/OsCTR2, reducing OsCTR2 phosphorylation and initiating downstream signaling. Our study unveils the dual role of MHZ3 in fine-tuning ethylene signaling activation, providing insights into the initial stages of the ethylene signaling cascade.

Unlike animals, plants have evolved to adapt to environmental changes in their fixed locations. Over millions of years of evolution, they have developed sophisticated signaling networks that allow rapid responses to stresses and facilitate growth recovery after stress events[1,2]. The plant-specific gaseous hormone, ethylene, has functioned as a signaling molecule predating land colonization[3], influencing various aspects of plant growth, development, and stress responses[4]. Serving as a pivotal regulator, ethylene signals through an intricate pathway to trigger biochemical and morphological changes in plants.

In Arabidopsis, ethylene is perceived by a group of endoplasmic reticulum (ER) membrane-integrated receptor proteins, including ETHYLENE RESISTANT 1 (ETR1), ETHYLENE RESPONSE SENSOR 1 (ERS1), ERS2, ETR2, and ETHYLENE INSENSITIVE 4 (EIN4)[5–8]. The

ethylene receptors exhibit both functional redundancy and divergent signaling capacities[9–11]. In air, the ethylene receptor/AtCTR1 complex is active, and AtCTR1 phosphorylates the positive regulator EIN2[12–14]. EIN2 is degraded by two F-box proteins, EIN2 TARGETING PROTEIN 1 (ETP1) and ETP2[15]. Upon exposure to ethylene, the ethylene receptor/AtCTR1 complex is inactive, and AtCTR1-mediated phosphorylation of EIN2 is inhibited[16]. The C-terminal end of EIN2 undergoes proteolytic cleavage and is transferred from the ER to the nucleus[17,18] or RNA processing body[19,20]. This translocation event effectively hinders the ubiquitination-mediated degradation process of ETHYLENE INSENSITIVE 3 (EIN3)/EIN3-LIKE1 (EIL1)[21] by EIN3 BINDING F-BOX1/2 (EBF1/2)[22–24], thereby exerting regulatory control over downstream ethylene-responsive genes. However, how ethylene binding would

[1]Key Lab of Seed Innovation, Institute of Genetics and Developmental Biology, Chinese Academy of Sciences, Beijing 100101, China. [2]College of Advanced Agricultural Sciences, University of Chinese Academy of Sciences, Beijing 100049, China. [3]Key Laboratory of Ministry of Education for Genetics, Breeding and Multiple Utilization of Crops, College of Agriculture, Fujian Agriculture and Forestry University, Fuzhou, China. [4]Guangdong Laboratory for Lingnan Modern Agriculture, College of Agriculture, South China Agricultural University, Guangzhou 510642, China. [5]The Sainsbury Laboratory, University of East Anglia, Norwich, UK. ✉e-mail: He.Zhao@tsl.ac.uk; ccyin@genetics.ac.cn; jszhang@genetics.ac.cn

trigger the switch of its receptors and AtCTR1 between active and inactive states is not yet fully understood. More research is needed to fully characterize the molecular changes governing this ethylene-induced transition.

In rice, mechanisms of ethylene signaling regulation have been identified, encompassing both conserved and divergent aspects when compared to Arabidopsis[25,26]. Conserved components include ethylene receptors OsERS1[25], OsERS2/MHZ12[27], OsETR2, and OsETR3[28]; Raf-like kinases OsCTR1/2[29,30]; MHZ7/OsEIN2[31]; MHZ6/OsEIL1, and OsEIL2[32]. Among the newly discovered components, MHZ3, a membrane-localized scaffold protein, interacts with the N-terminal Nramp-like domain of OsEIN2 to prevent its ubiquitination-mediated degradation[33]. MHZ11 encodes a GDSL lipase that affects membrane sterol content, thereby influencing the interaction between ethylene receptors and OsCTR2[30]. MHZ1/OsHK1 is a histidine kinase, and the ethylene signaling mediated by MHZ1 is independent of the traditional OsEIN2-mediated pathway[27]. Additionally, the translational regulator MHZ9 directly binds to the 3′UTR of *EBF1/2* mRNA, suppressing their translation, thus positively regulating ethylene signaling[34]. Cross-talks between ethylene and other hormones, such as auxin, ABA, and JA, were also identified at new levels[35–39]. Rice constitutes a complementary monocot model, facilitating the exploration of both conserved and specialized aspects of the ethylene signaling pathway. These findings indicate the existence of unidentified components or regulatory mechanisms within the ethylene signaling cascade.

We previously demonstrated that OsCTR2 exists in both phosphorylated (OsCTR2-P) and non-phosphorylated forms (OsCTR2), and ethylene exposure triggers a shift towards the latter[30]. However, the mechanism of how ethylene triggers changes in OsCTR2 phosphorylation remains elusive. Here, we unveil a significant role of MHZ3 in ethylene signaling, extending beyond its function as a stabilizer for the key positive regulator OsEIN2 in signal transduction. MHZ3 dynamically associates with ethylene receptors to facilitate ethylene receptors-OsCTR2 interaction to maintain OsCTR2 phosphorylation and activity. Ethylene exposure alters these binding dynamics, attenuating MHZ3-receptor connections to suppress OsCTR2 activity and trigger downstream signaling. Our investigation elucidates a critical regulatory mechanism within the signaling complex comprising MHZ3, ethylene receptors, and OsCTR2, broadening our understanding of the early events of ethylene signaling.

## Result

### The membrane protein MHZ3 is required for OsCTR2 phosphorylation

We previously demonstrated that ethylene treatment converts OsCTR2 from a phosphorylated form to a non-phosphorylated form[30]. Further investigation into this phosphorylation dynamics showed that the intensity of the OsCTR2-P band weakens gradually during prolonged exposure to ethylene, whereas the non-phosphorylated form strengthens, evident within 15 min (Fig. 1a). Upon ethylene removal, the OsCTR2 phosphorylation recovered rapidly within 15 min (Fig. 1b). Expression patterns of ethylene-responsive genes mirror OsCTR2 phosphorylation dynamics (Supplementary Fig. 1a, b). In Arabidopsis, autophosphorylation of AtCTR1 is crucial for its kinase activity and homodimer formation[40,41]. Here, we investigated the effect of autophosphorylation on the function of OsCTR2 in rice. By aligning the amino acid sequence with AtCTR1, we introduced mutations in OsCTR2 at the corresponding residues (T665/S668/T671), replacing them with Ala to generate the phosphor-dead mutant (OsCTR2^AAA) (Supplementary Fig. 2a). In both rice wild-type (WT) and *Osctr2* protoplasts, OsCTR2^D-E (kinase-dead) and OsCTR2^AAA (phosphor-dead) showed no detectable phosphorylation compared to wild-type OsCTR2, losing their inhibitory effect on ethylene-responsive genes (Supplementary Fig. 2b, c). These results indicate that the autophosphorylation of OsCTR2 is essential for its function, and the rapid

phosphorylation changes of OsCTR2 upon ethylene exposure reflect the switch on/off of ethylene signaling cascade, serving as a sensitive indicator enabling capture of pathway activation and attenuation.

We investigated which components of the rice ethylene signaling pathway may affect the OsCTR2 phosphorylation and found that, while the ethylene receptor OsERS2 gain-of-function mutant *Osers2^d* and the GDSL lipase mutant *mhz11* locked the OsCTR2 in the phosphorylation state (Supplementary Fig. 3a, b)[30], the *Osein2-1* and *Oseil1* mutants exhibited OsCTR2 phosphorylation dynamics similar to wild-type (Fig. 1c). This is consistent with the current ethylene signaling model, of which the ethylene receptors and MHZ11 work upstream of OsCTR2, while OsEIN2 and OsEIL1 downstream of OsCTR2 for function[26]. It is interesting to note that *mhz3-1* mutant completely abolished OsCTR2 phosphorylation under air conditions or upon treatment with ethylene and 1-Methylcyclopropene (1-MCP), which is an ethylene receptor antagonist that inhibits endogenous ethylene perception (Fig. 1c and Supplementary Figs. 4b, 5a, b). Generally, the 1-MCP treatment would maintain the OsCTR2 phosphorylation state in WT and downstream mutants (Fig. 1c). These results suggest that MHZ3 is likely involved in the phosphorylation of OsCTR2 to regulate ethylene signaling initiation. Intriguingly, MHZ3 was previously identified as a stabilizer of OsEIN2 that acts downstream of OsCTR2[33], suggesting multiple functions of MHZ3.

To validate that *MHZ3* gene is responsible for the altered OsCTR2 phosphorylation in *mhz3-1* mutant, we examined the OsCTR2 phosphorylation state in the complemented line *mhz3-1/gMHZ3* (Fig. 1d). Expression of *MHZ3* genomic DNA in *mhz3-1* plants rescued basal OsCTR2 phosphorylation in air, while ethylene retained capacity to trigger phospho-reduction (Fig. 1e). Four additional *mhz3* mutant alleles exhibited similarly disrupted OsCTR2 phosphorylation in air (Fig. 1f, g), suggesting that the loss of MHZ3 function is responsible for the attenuated OsCTR2 phosphorylation. The 1-MCP treatment partially recovered the OsCTR2 phosphorylation in *mhz3-2*, *mhz3-3*, and *mhz3-4*, suggesting that these three alleles may have the residual weak function (Fig. 1g). These results indicate that MHZ3 is required for the OsCTR2 phosphorylation.

We further tested the effect of MHZ3 on OsCTR2 phosphorylation by examining the OsCTR2 phosphorylation status in *mhz3*/MHZ3-GFP and MHZ3-overexpression (MHZ3-OE) transgenic plants. Expression of MHZ3 protein was confirmed in all these transgenic lines by using MHZ3-specific antibody (Fig. 1h). Transgenic lines with higher MHZ3 accumulation displayed additional, higher molecular weight OsCTR2 phospho-bands, whereas line MHZ3-OE#26 with a lower MHZ3 level had only low molecular weight OsCTR2-P band (Fig. 1i). All these slower-migrating bands disappeared upon λ-protein phosphatase (λ-PPase) treatment (Fig. 1i, j), confirming the additional phosphorylation of OsCTR2 in the MHZ3-overexpressing lines. Ethylene-triggered OsCTR2 dephosphorylation exhibits delayed kinetics in transgenic plants with high MHZ3 expression (Fig. 1i). These results indicate that elevated MHZ3 level promotes OsCTR2 phosphorylation and at least partially counteracts ethylene-triggered phosphor-shift of OsCTR2.

### Ethylene receptors differentially regulate OsCTR2 phosphorylation

We previously demonstrated that rice ethylene receptor OsERS2 interacts with OsCTR2, and OsERS2 gain-of-function mutant *Osers2^d* locked OsCTR2 at the phosphorylation status[30]. To determine the predominant ethylene receptor(s) governing OsCTR2 activity, we examined multiple receptor loss-of-function mutants for impacts on OsCTR2 phosphorylation status. The ethylene receptor single mutants *Osers1*, *Osers2*, *Osetr2*, and *Osetr3*, as well as double mutants *Osetr2 ers2* and *Osetr2 etr3* were examined (Fig. 2a). Although these ethylene receptor mutants were generated in the Dongjin (DJ) background[32], OsCTR2 phosphorylation response patterns were similar between DJ and Nipponbare (WT) (Supplementary Fig. 6). In air-grown *Osetr2*, *Osetr2 ers2*, and *Osetr2 etr3* mutants, the

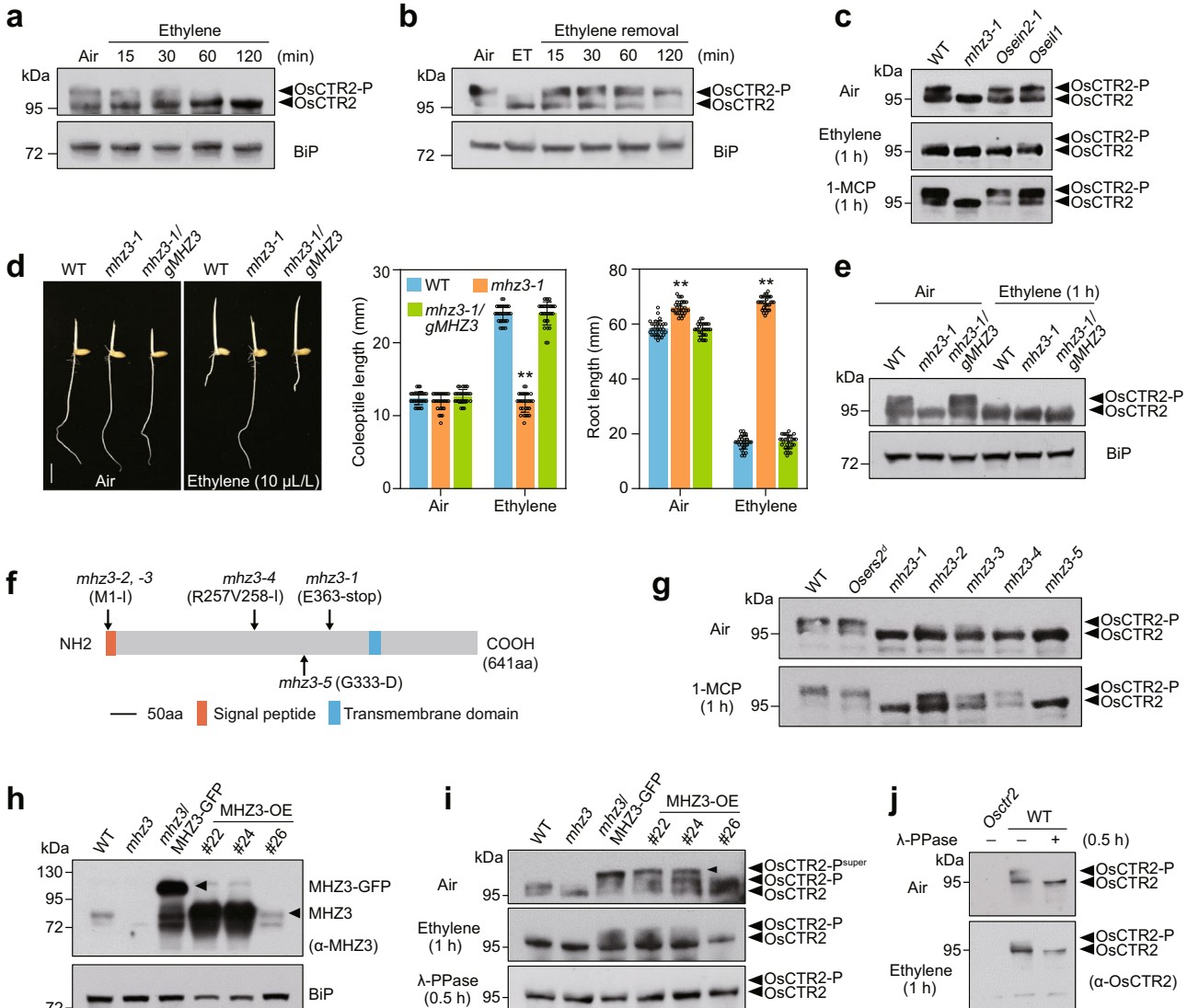

**Fig. 1 | MHZ3 is required and sufficient for OsCTR2 phosphorylation. a** The phosphorylation of OsCTR2 decreases in response to ethylene. Etiolated seedlings of the wild-type (WT) were exposed to 10 μL/L ethylene for varying durations. **b** OsCTR2 phosphorylation is recovered after the removal of ethylene (ET). Etiolated seedlings of the WT were treated with 10 μL/L ethylene for 120 min, followed by sampling at different time points after ethylene removal. **c** The phosphorylation state of OsCTR2 in *mhz3-1* is compared with that in WT, *Osein2-1* and *Oseil1*. Etiolated seedlings were treated with air, 10 μL/L ethylene, and 10 μL/L 1-Methylcyclopropene (1-MCP) for 1 hour. **d** Functional complementation of *mhz3-1* was achieved by transforming the mutant with *MHZ3* genomic DNA. Etiolated seedlings grown for 3 days with or without 10 μL/L ethylene are shown, scale bars = 10 mm. Coleoptile and root lengths are means ± SD, *n* = 30 biologically independent plants (\*\**P* < 0.01; two-tailed Student's *t*-test; compared with WT). **e** OsCTR2 phosphorylation is rescued in *mhz3-1/gMHZ3* plants. Two-day-old etiolated seedlings were treated with or without 10 μL/L ethylene for 1 hour. **f** Schematic diagrams of the mutation sites of the allelic mutants of MHZ3. **g** The OsCTR2

phosphorylation state in *mhz3* allelic mutants. Total proteins were extracted from 2-day-old etiolated seedlings of WT, *Osers2^d* (OsERS2 gain-of-function mutation), and *mhz3* allelic mutants. **h** Overexpression of MHZ3 (MHZ3-OE) in various transgenic plants. MHZ3's protein content was evaluated utilizing the anti-MHZ3 antibody. **i** MHZ3 enhances the phosphorylation of OsCTR2. Two-day-old etiolated seedlings were treated with 10 μL/L ethylene for 1 hour, while air-treated seedlings' protein extracts were treated with λ-Protein Phosphatase (λ-PPase) for 0.5 hours. **j** λ-PPase treatment resulted in the removal of OsCTR2 phosphorylation. OsCTR2-P denotes the phosphorylated OsCTR2 protein, while OsCTR2 represents the non-phosphorylated form. Additionally, OsCTR2-P^super indicates further phosphorylation modifications in MHZ3-OE plants. OsCTR2 phosphorylation was detected using anti-OsCTR2 antibody, with Binding Immunoglobulin Protein (BiP) as the ER membrane marker for internal reference. Three independent experiments were repeated with similar results. Uncropped blots and source data are in the Source Data file.

phosphorylated OsCTR2 was nearly undetectable. While 1-MCP treatment partially recovered phospho-levels in *Osetr2* plants, it failed to rescue phosphorylation in the double mutants (Fig. 2a). In contrast, under air, ethylene, or 1-MCP treatment, OsCTR2 phosphorylation profiles in *Osers1*, *Osers2*, and *Osetr3* mutants resembled WT (Fig. 2a). Notably, ethylene receptor mutations did not affect MHZ3 accumulation (Supplementary Fig. 7a, b). These data reveal a critical role for ethylene receptors, particularly OsETR2, in regulating OsCTR2 phosphorylation.

Considering that the ethylene receptors interact with and affect OsCTR2 phosphorylation[30], and typically localize to the ER membrane[4], we investigated whether ethylene receptor mutations would affect the subcellular localization of OsCTR2. We extracted the total proteins from WT, *Osetr2*, *Osetr2 ers2*, and *Osetr2 etr3* and separated them into soluble and microsomal fractions by ultracentrifugation. Compared to the WT, the microsomal fractions of *Osetr2*, *Osetr2 ers2*, and *Osetr2 etr3* mutants displayed significantly reduced levels of OsCTR2, while the cytoplasmic fraction showed relatively higher levels

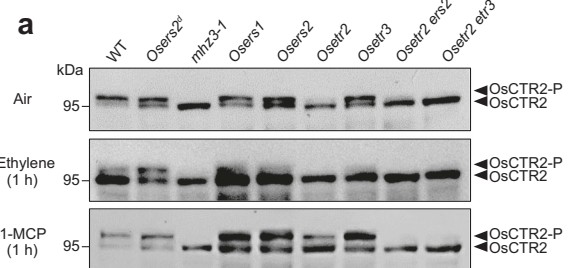

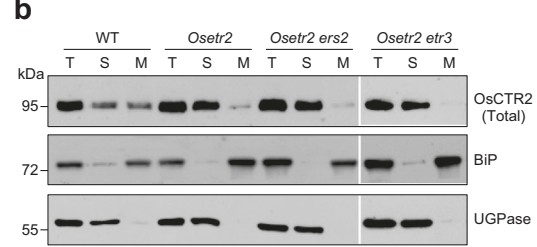

**Fig. 2 | Ethylene receptors differentially regulate OsCTR2 phosphorylation.**
**a** the phosphorylation state of OsCTR2 in various ethylene receptor mutants. Two-day-old etiolated seedlings were treated with air, 10 μL/L ethylene, or 10 μL/L 1-MCP for 1 hour, respectively. Protein extracts were analyzed using the anti-OsCTR2 antibody to assess the phosphorylation state of OsCTR2. Notably, *Osers2$^d$* is characterized by a dominant mutation leading to the gain of function of OsERS2, whereas the other ethylene receptor mutants exhibit loss-of-function mutations.

**b** Membrane association of OsCTR2 in WT, *Osetr2*, *Osetr2 ers2*, and *Osetr2 etr3*. Equal amounts of total (T), soluble (S), and microsomal membranes (M) proteins were subjected to immunoblotting for OsCTR2, BiP (an ER membrane marker), and UDP-glucose pyrophosphorylase (UGPase, a cytoplasmic marker). Three independent experiments were repeated with similar results. Uncropped blots in Source Data file.

of OsCTR2 proteins (Fig. 2b). Different from Arabidopsis AtCTR1, OsCTR2 was not detectable in the nucleus of *Osetr2*, *Osetr2 ers2*, or *Osetr2 etr3* plants (Supplementary Fig. 8), implying possible functional differentiation. These results indicate that ethylene receptors are required for the proper membrane localization of OsCTR2.

### MHZ3 physically interacts with OsETR2 and OsERS2
To further investigate the relationship between MHZ3 and ethylene receptors, we selected OsERS2 from subfamily I and OsETR2 from subfamily II as representatives of their respective subfamilies. Given that both MHZ3 and the ethylene receptors are required for OsCTR2 phosphorylation and that MHZ3 is also localized to the ER membrane[33], we investigated whether MHZ3 has a direct relationship with ethylene receptors. Firstly, we found that MHZ3 co-localizes to the ER membrane with OsETR2 and OsERS2 in *Nicotiana benthamiana* leaves and rice protoplasts (Fig. 3a and Supplementary Fig. 9). We then employed the split-ubiquitin membrane yeast two-hybrid system (MbYTH)[42] to examine the interaction between MHZ3 and OsETR2, as well as OsERS2. Compared to the negative controls, yeast cells co-expressing MHZ3-Cub and NubG-OsETR2 or NubG-OsERS2 were able to grow on selective media (-LWH) (Fig. 3b), indicating a direct interaction between MHZ3 and both OsETR2 and OsERS2 in yeast cells. To validate the interactions in vivo, we conducted co-immunoprecipitation (Co-IP) assays in rice protoplasts. Plasmids encoding *MHZ3-GFP* and *OsETR2-Myc*, or *MHZ3-GFP* and *OsERS2-Myc* were co-transfected into *Osetr2 mhz3* or *Osers2 mhz3* protoplasts, respectively. GFP served as the negative control. MHZ3-GFP co-immunoprecipitated efficiently with both OsETR2-Myc and OsERS2-Myc compared to the GFP control (Fig. 3c). We also carried out the luciferase complementation imaging (LCI) assays[43] to further validate the interactions. Robust luciferase activity was observed when co-expressing Cluc-OsETR2 with MHZ3-Nluc, or Cluc-OsERS2 with MHZ3-Nluc, whereas no signal was detected in the control groups (Fig. 3d). All these results demonstrate that MHZ3 interacts with both OsETR2 and OsERS2 in vitro and in vivo.

Given that the ethylene receptors are multiple-domain proteins, we sought to determine whether MHZ3 interacts with any specific domains of OsETR2 and OsERS2 using Co-IP in rice protoplasts. Utilizing the 3D structures predicted by AlphaFold2[44] and membrane-embedding information provided by the TmAlphaFold (https://tmalphafold.ttk.hu/)[45], we determined the boundaries of each domain of OsETR2 and OsERS2 (Supplementary Fig. 10). Subsequently, we cloned the coding sequences of each domain into an expression vector containing the nYFP-FLAG tag. Co-transfection with *MHZ3-GFP*

or *GFP* was then conducted in rice protoplasts *Osetr2 mhz3* and *Osers2 mhz3*. The results showed that OsETR2-TMGAF, OsETR2-GAFHR, and OsERS2-TMGAF were co-immunoprecipitated with MHZ3-GFP (Fig. 3e, f). These results support that MHZ3 interacts with both ethylene receptors OsETR2 and OsERS2, possibly through the trans-membrane region and GAF domain.

### MHZ3 and ethylene receptors collaborate to regulate the phosphorylation of OsCTR2
Given that both MHZ3 and ethylene receptors are required for basal OsCTR2 phosphorylation, and MHZ3 interacts with ethylene receptors, we conducted an epistatic analysis to determine the genetic relationship between MHZ3 and ethylene receptors in OsCTR2 phosphorylation regulation. *Osers2$^d$ mhz3-1* double mutant was generated by crossing *Osers2$^d$* and *mhz3-1* single mutants (Supplementary Fig. 11a) and checked for OsCTR2 phosphorylation. Under air, ethylene, or 1-MCP treatment, OsCTR2 phosphorylation patterns in the *Osers2$^d$ mhz3-1* double mutant were similar to that in the *mhz3-1* single mutant (Fig. 4a). This genetic analysis positions MHZ3 downstream of or epistatic to the ethylene receptors in modulating OsCTR2 phosphorylation, suggesting that ethylene receptor signaling depends on functional MHZ3.

We also generated MHZ3-GFP/*Osetr2* homozygous line through crossing (Supplementary Fig. 11b). The elevated OsCTR2 phosphorylation induced by MHZ3-GFP overexpression was abolished upon introduction of *Osetr2* mutation especially in air (Fig. 4b). Moreover, ethylene-triggered OsCTR2 phospho-shift in MHZ3-GFP/*Osetr2* resembled the pattern in *Osetr2* single mutant (Fig. 4b). This result indicates that MHZ3-facilitated OsCTR2 phosphorylation requires intact ethylene receptor activity.

To further investigate the relationship between ethylene receptors and MHZ3 on regulating OsCTR2 phosphorylation, we constructed *35 S:OsETR2-Myc* and *35 S:OsERS2-Myc* stable transgenic plants in WT and *mhz3* backgrounds to check the phosphorylation state of OsCTR2 (Supplementary Figs. 12–14). Both OsETR2 and OsERS2 promoted OsCTR2 phosphorylation in the presence of MHZ3, however, when MHZ3 function is disrupted, the ethylene receptor-mediated maintenance of OsCTR2 phosphorylation is abolished (Fig. 4c, d). These results further suggest that functional MHZ3 is required by both OsETR2 and OsERS2 to maintain OsCTR2 phosphorylation.

It should be noted that ethylene insensitivity and locked OsCTR2 phosphorylation observed in Osers2$^d$-Myc expressing plants demonstrate that the fused Myc tag does not impact ethylene receptor function (Supplementary Fig. 15a–d). Overexpression of OsETR2 and

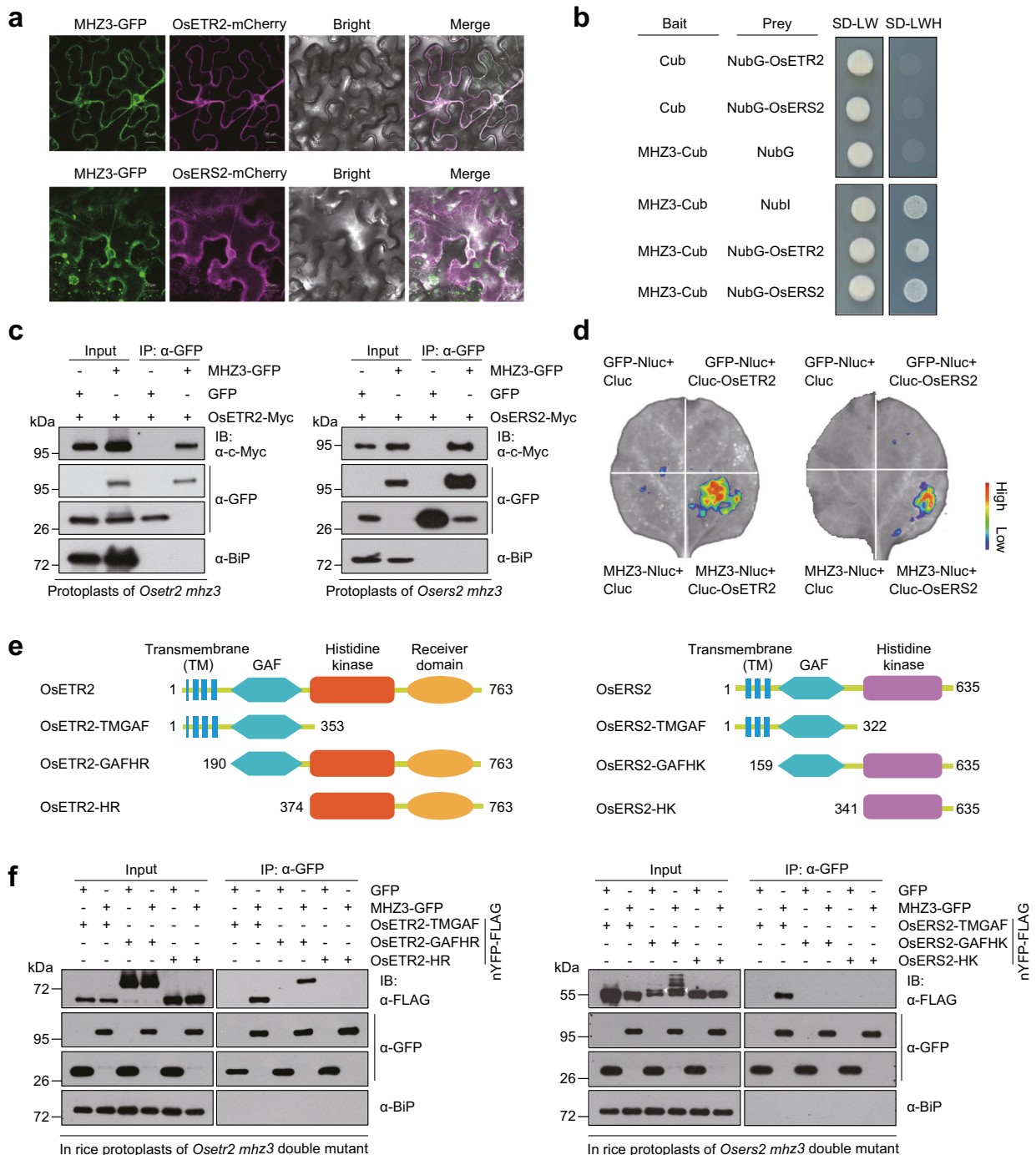

**Fig. 3 | MHZ3 physically interacts with OsETR2 and OsERS2. a** MHZ3 co-localizes with OsETR2 and OsERS2 in *N. benthamiana* leaves. Scale bars = 20 μm. **b** Split-ubiquitin membrane yeast two-hybrid assay for interaction of MHZ3 with OsETR2 and OsERS2. Cub, C-terminal half of the ubiquitin. NubI, N-terminal half of ubiquitin. NubG, Substitution of Ile-13 in wild-type NubI with glycine reduces Nub and Cub affinity. **c** Co-immunoprecipitation (Co-IP) assays for interaction of MHZ3 with OsETR2 and OsERS2. The indicated constructs were co-transformed into *Osetr2 mhz3* or *Osers2 mhz3* rice protoplasts. The GFP was used as a negative control. Total proteins were immunoprecipitated with GFP Trap and immunoblotted with anti-GFP, anti-c-Myc, and anti-BiP antibodies. IP: Immunoprecipitation. IB: Immunoblotting. **d** Luciferase complementation imaging assays for the interaction of MHZ3 with OsETR2 and OsERS2. Luciferase activity is visualized with artificial color from

low (purple) to high (red). Nluc, N-terminal portion of firefly luciferase; Cluc, C-terminal portion of firefly luciferase. **e** Diagrams of full-length and truncated versions of OsETR2 and OsERS2 used in interaction domain mapping studies. GAF is short for cGMP-specific phosphodiesterases, Adenylyl cyclases, and FhlA. HR stands for histidine kinase domain plus receiver domain. HK stands for histidine kinase domain. **f** Co-IP assays for mapping interaction domains of OsETR2 and OsERS2 associated with MHZ3. The GFP was used as a negative control. The constructs were co-transformed into *Osetr2 mhz3* or *Osers2 mhz3* rice protoplasts. Total proteins were immunoprecipitated with GFP Trap and immunoblotted with anti-GFP, anti-FLAG, and anti-BiP antibodies. The triangle indicates the band for OsERS2-GAFHK-nYFP-FLAG. Two independent experiments were repeated with similar results. Uncropped blots in Source Data file.

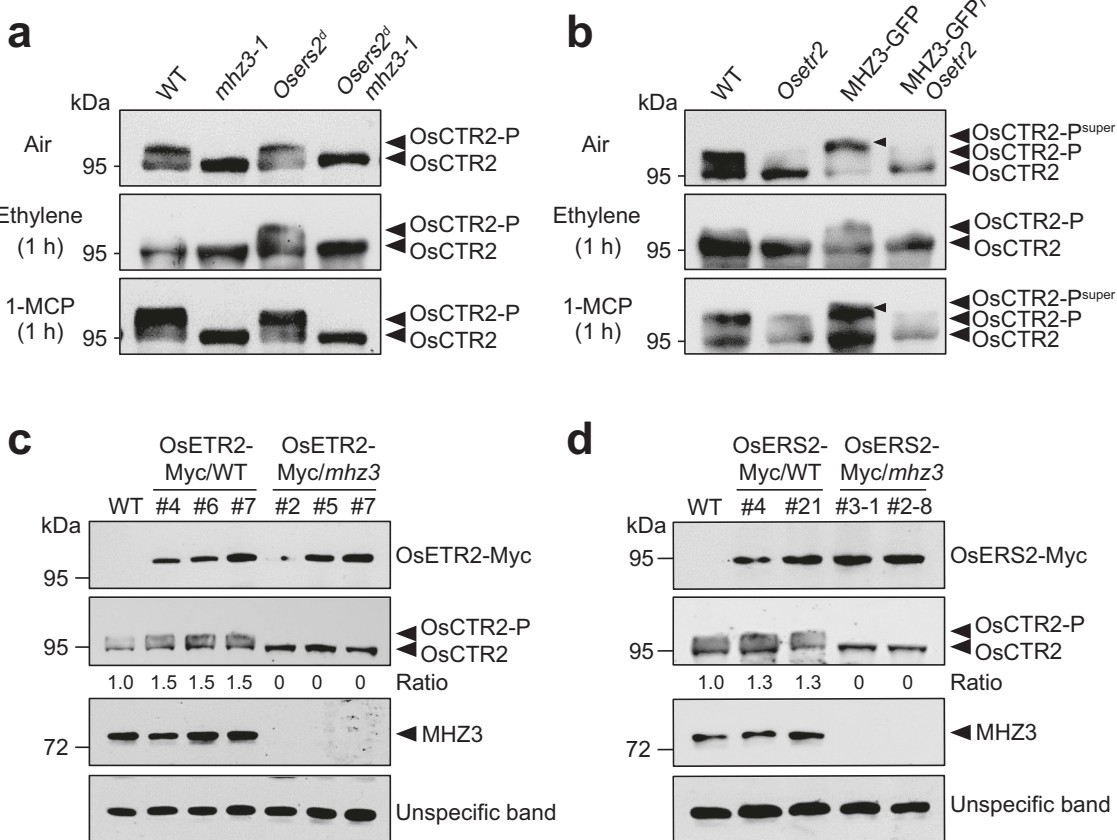

**Fig. 4 | MHZ3 and ethylene receptors work collaboratively to regulate the phosphorylation of OsCTR2. a** The phosphorylation state of OsCTR2 in *Osers2^d mhz3-1* is similar to *mhz3-1*. *Osers2^d* is a gain-of-function mutant of OsERS2. **b** The phosphorylation state of OsCTR2 in MHZ3-GFP/*Osetr2* is similar to *Osetr2*. Small black arrows indicate OsCTR2-P^super. For **a** and **b**, 2-day-old etiolated seedlings were treated with air, 10 μL/L ethylene, or 10 μL/L 1-MCP for 1 hour before sampling. **c** OsETR2's elevation of OsCTR2 phosphorylation requires MHZ3. **d** OsERS2's elevation of OsCTR2 phosphorylation requires MHZ3. For **c** and **d**, the MHZ3 mutation does not affect the protein levels of OsETR2 and OsERS2. Total protein was extracted from 2-day-old etiolated seedlings. The protein content of OsETR2 was determined using an anti-c-Myc antibody, while the phosphorylation state of OsCTR2 was assessed using an anti-OsCTR2 antibody. An unspecific band was utilized as an internal reference. Ratio represents the ratio of OsCTR2-P to OsCTR2, which is set to 1.0 in the wild type (WT). Three independent experiments were repeated with similar results. Uncropped blots in Source Data file.

OsERS2 in WT resulted in reduced ethylene sensitivity in terms of coleoptile and root growth, consistent with our previous result[28]. Overexpression of OsETR2 and OsERS2 in *mhz3* background showed an ethylene-insensitive phenotype, similar to that observed in *mhz3* (Supplementary Figs. 12, 13). Consistent with the finding that ethylene receptor mutation does not influence MHZ3 accumulation, the mutation of MHZ3 similarly did not impact the protein abundance of both ethylene receptors, indicating that the relationship between MHZ3 and the ethylene receptors extends beyond the regulation of protein accumulation (Fig. 4c, d and Supplementary Fig. 16).

### MHZ3 enhances the interaction between ethylene receptors and OsCTR2

Given the physical association between MHZ3 and the ethylene receptors, and their genetic relationship in controlling OsCTR2 phosphorylation, we hypothesized that MHZ3 may affect the ethylene receptors-OsCTR2 interaction by affecting the recruitment of OsCTR2 by ethylene receptors. To test this hypothesis, we selected OsETR2-Myc/WT#7 and OsETR2-Myc/*mhz3*#5, and OsERS2-Myc/WT#21 and OsERS2-Myc/*mhz3*#3-1 transgenic lines with comparable receptor gene expression levels to examine the effect of MHZ3 on the interaction between ethylene receptors and OsCTR2 by Co-IP assays. Co-IP results showed that interactions between ethylene receptors OsETR2, OsERS2, and OsCTR2 were stronger in WT than those in *mhz3* (Fig. 5a, b). Intriguingly, the phosphorylated form of OsCTR2 was preferentially enriched in ethylene receptor immunoprecipitates compared with the input samples. These results indicate that most likely the ethylene receptors associate with phosphorylated OsCTR2 for receptor signaling.

To investigate the impact of MHZ3 on the membrane localization of OsCTR2, we utilized the membrane protein isolation assay to examine the localization of OsCTR2. Compared to WT, membrane-localized OsCTR2 is significantly reduced in *mhz3-1* (Fig. 5c). These results indicate that MHZ3 is required for recruitment of OsCTR2 to the ER membrane probably mediated by ethylene receptors.

To test whether MHZ3 can strengthen the interaction between ethylene receptors and OsCTR2 in plants, luciferase complementation imaging assays were carried out. Agrobacterium carrying *OsETR2* or *OsERS2* with the Cluc tag was mixed with Agrobacterium carrying *OsCTR2* with the Nluc tag. Subsequently, the mixture was co-infiltrated with either GFP or MHZ3-GFP into leaves of similar size, and fluorescence intensity was observed. Results showed that MHZ3 enhanced the interaction between OsETR2 and OsERS2 with OsCTR2 without significantly affecting the protein accumulation of OsETR2, OsERS2, and OsCTR2 (Fig. 5d, e). To further confirm the results, a semi-in vitro pull-down (PD) assay was performed. Equal amounts of ethylene

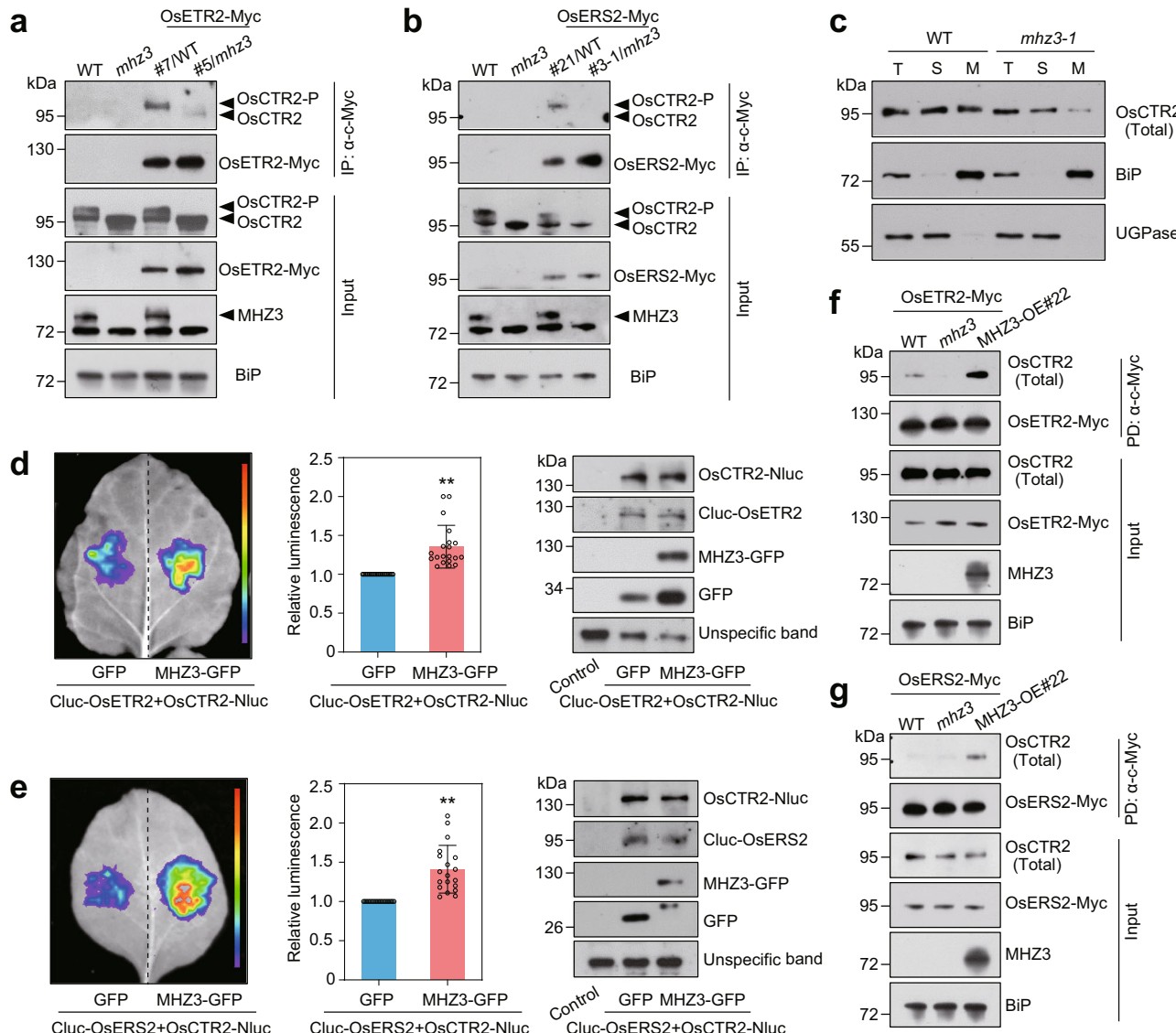

**Fig. 5 | MHZ3 is required for the interaction between the ethylene receptors and OsCTR2. a, b** Co-IP assays indicate that MHZ3 mutation disrupted the inter-action between OsCTR2 and OsETR2 (**a**) or OsERS2 (**b**). Transgenic rice seedlings stably expressing OsETR2-Myc/WT, OsETR2-Myc/*mhz3*, OsERS2-Myc/WT, and OsERS2-Myc/*mhz3* were grown in the dark for 2 days. Total proteins were immu-noprecipitated with anti-c-Myc affinity gel and immunoblotted with anti-c-Myc, anti-OsCTR2, anti-MHZ3, and anti-BiP antibodies. WT and *mhz3* were used as negative controls. **c** Membrane association of OsCTR2 in WT and *mhz3-1*. Equal amounts of total protein (T), soluble protein (S), and microsomal membranes (M) were immunoblotted for OsCTR2, BiP (ER membrane marker), and UGPase (cyto-plasm marker). **d, e** Luciferase complementation imaging assays demonstrate that MHZ3 enhances the interaction between OsCTR2 and OsETR2 (**d**) or OsERS2 (**e**). An equal amount of the Agrobacteria harboring *Cluc-OsETR2* or *Cluc-OsERS2* expres-sing vectors plus *OsCTR2-Nluc* vector was co-infiltrated into *N. benthamiana* leaves.

The Agrobacteria harboring *GFP* or *MHZ3-GFP* vectors were introduced to compare the effects. The relative luminescence values represent the means ± SD, with n = 20 (**d**) and n = 19 (**e**) biologically independent samples (\*\*P < 0.01, two-tailed Student's *t*-test; compared with GFP). Immunoblot analysis showed similar expression levels of Cluc-OsETR2, Cluc-OsERS2, and OsCTR2-Nluc proteins in each experimental group. **f, g** MHZ3 enhances interactions of OsETR2 (**f**) and OsERS2 (**g**) with OsCTR2 in semi-in vitro pull-down (PD) assays. Equal amounts of OsETR2-Myc and OsERS2-Myc protein purified from OsETR2-Myc/*mhz3* and OsERS2-Myc/*mhz3* plants are added to protein homogenates from WT, *mhz3*, and MHZ3-OE#22. Total homo-genate proteins were immunoprecipitated with anti-c-Myc affinity gel and immu-noblotted with anti-c-Myc, anti-OsCTR2, anti-MHZ3, and anti-BiP antibodies. Three independent experiments were repeated with similar results. Uncropped blots and source data are in the Source Data file.

receptor proteins purified from OsETR2-Myc/*mhz3*#7 and OsERS2-Myc/*mhz3*#3-1 were incubated with protein crude extracts from WT, *mhz3*, and MHZ3-OE#22. OsETR2-Myc and OsERS2-Myc bound more OsCTR2 in MHZ3-OE#22 than those in the WT and *mhz3* (Fig. 5f, g), suggesting that MHZ3 enhances the binding of OsETR2 and OsERS2 to OsCTR2. All these findings indicate that MHZ3 facilitates the interac-tion between OsETR2 and OsERS2 with OsCTR2, potentially modulat-ing ethylene-induced phosphorylation changes in OsCTR2 by regulating this interaction.

## Ethylene attenuates the interaction between the ethylene receptors and OsCTR2, as well as MHZ3

We demonstrated that MHZ3 and ethylene receptors are required for OsCTR2 phosphorylation and ER membrane localization, but how ethylene triggers OsCTR2 phosphorylation changes through ethylene receptors and MHZ3 remains unclear. In Arabidopsis, ethylene triggers the subcellular trafficking of AtCTR1[41]. We speculate that ethylene binding would weaken the interaction between ethylene receptors and MHZ3, altering OsCTR2 subcellular localization and changing the

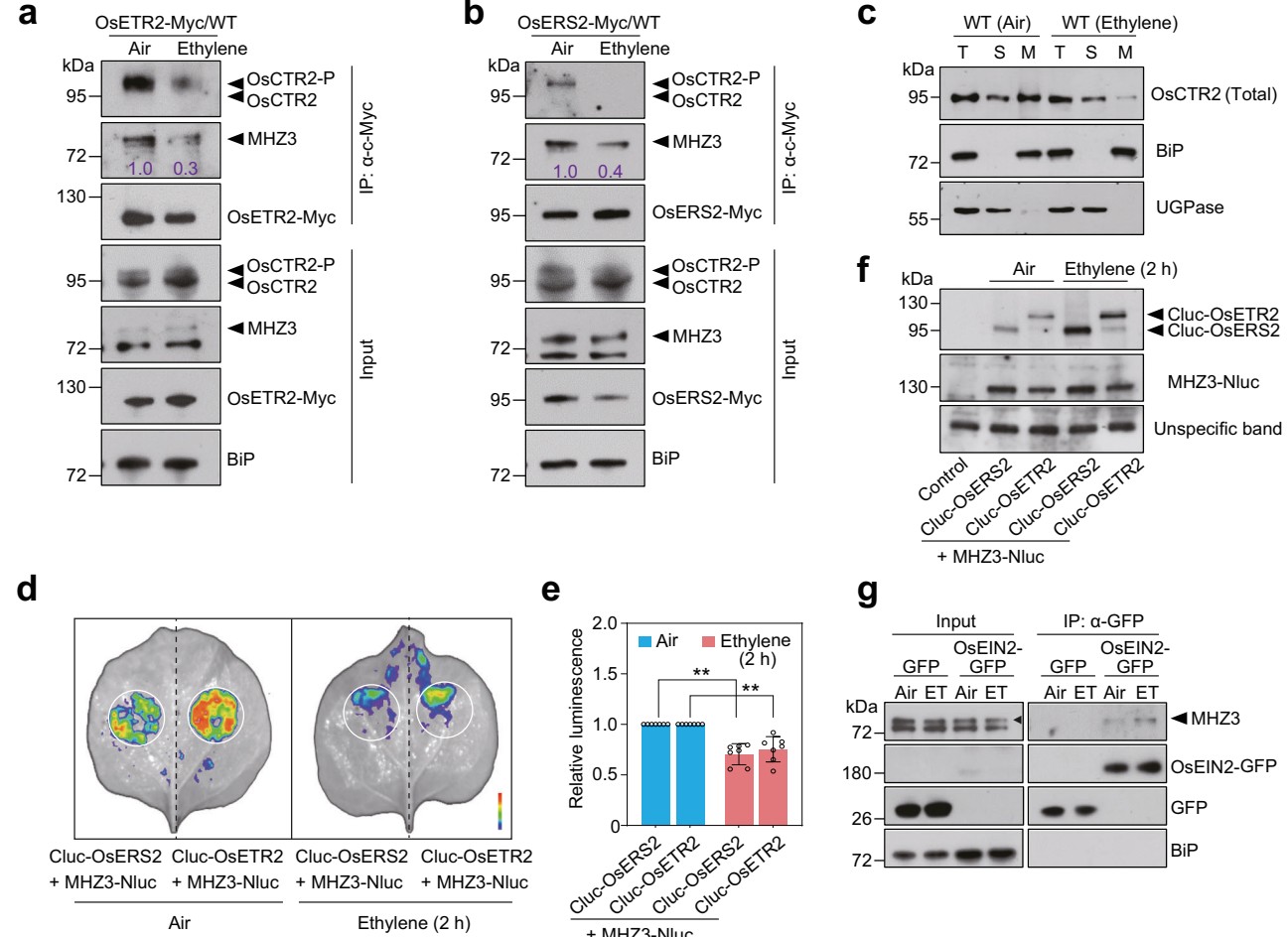

**Fig. 6 | Ethylene attenuates the interaction between the ethylene receptors and OsCTR2, as well as MHZ3. a, b** Co-IP assays indicate that ethylene impaired the interaction between OsETR2 (**a**) or OsERS2 (**b**) and OsCTR2, and weakened the interaction between MHZ3 and OsETR2 (**a**) or OsERS2 (**b**). Transgenic rice seedlings stably expressing OsETR2-Myc/WT and OsERS2-Myc/WT were grown in the dark for 2 days, followed by a 1-hour treatment with 10 μL/L ethylene. Total proteins were immunoprecipitated with anti-c-Myc affinity gel and immunoblotted with anti-c-Myc, anti-OsCTR2, anti-MHZ3, and anti-BiP antibodies. **c** Membrane association of OsCTR2 after 10 μL/L ethylene treatment in WT. Equal amounts of total protein (T), soluble protein (S), and microsomal membranes (M) were immunoblotted for OsCTR2, BiP (ER membrane marker), and UGPase (cytoplasm marker). **d** Luciferase complementation imaging assays indicate that ethylene attenuated the interaction between OsETR2 or OsERS2 and MHZ3. Equal amounts of the Agrobacteria harboring *MHZ3-Nluc* vector were co-infiltrated with Agrobacteria harboring either the *Cluc-OsETR2* or *Cluc-OsERS2* vectors into *N. benthamiana* leaves. After 48 hours at 22 °C, some plants underwent a 2-hour treatment with 10 μL/L ethylene, while others remained in ambient air. **e** The statistical analysis of the luminescence intensity in (**d**). The values represent the means ± SD, *n* = 7 biologically independent samples (**P < 0.01; two-tailed Student's *t*-test; compared to the corresponding Air). **f** Immunoblot analysis for expression levels of Cluc-OsETR2, Cluc-OsERS2, and MHZ3-Nluc proteins in each experimental group. An unspecific band was utilized as an internal reference. **g** Ethylene (ET) promotes the interaction between OsEIN2 and MHZ3. Transgenic rice seedlings stably expressing OsEIN2-GFP were grown in the dark for 2 days, followed by a 1-hour treatment with 10 μL/L ethylene. Each experiment was repeated at least three times with similar results. Three independent experiments were repeated with similar results. Uncropped blots and source data are in the Source Data file.

OsCTR2 phosphorylation state. To test this model, we conducted Co-IP assays probing for ethylene receptor-bound OsCTR2 and MHZ3 using OsETR2-Myc/WT or OsERS2-Myc/WT seedlings treated with or without ethylene (Fig. 6a, b). Ethylene treatment reduced co-purification of both OsCTR2 and MHZ3 by the ethylene receptors-Myc proteins (Fig. 6a, b). These results demonstrate that ethylene perception through its receptors attenuates physical associations between MHZ3 and the receptors, as well as interactions between the receptors and OsCTR2.

We performed a membrane protein isolation assay to examine the membrane association of OsCTR2 in WT with or without ethylene treatment (Fig. 6c). The membrane association of OsCTR2 was reduced upon ethylene treatment compared with that in air. To further validate the effect of ethylene on the interaction between MHZ3 and ethylene receptors, *MHZ3-Nluc* was co-infiltrated with *Cluc-OsERS2* or

*Cluc-OsERS2* into *Nicotiana benthamiana* leaves of similar size. Leaves treated with or without ethylene were observed in the same field of view (Fig. 6d, e). Results also demonstrated that ethylene inhibited the interaction between MHZ3 and OsETR2 or OsERS2, without decreasing the protein levels (Fig. 6d–f). Taken together, our results imply that ethylene weakens the interaction between the ethylene receptors and MHZ3, and OsCTR2, and reduces the membrane association of OsCTR2.

Considering that ethylene attenuates MHZ3 binding to the ethylene receptors, and our previous study demonstrates that MHZ3 associates with OsEIN2[33], we evaluated how ethylene influences the MHZ3-OsEIN2 interaction. Co-IP results revealed that OsEIN2 showed stronger interaction with MHZ3 in the presence of ethylene than in air (Fig. 6g). This result demonstrates that ethylene stimulates the interaction between MHZ3 and OsEIN2 to facilitate signal transduction.

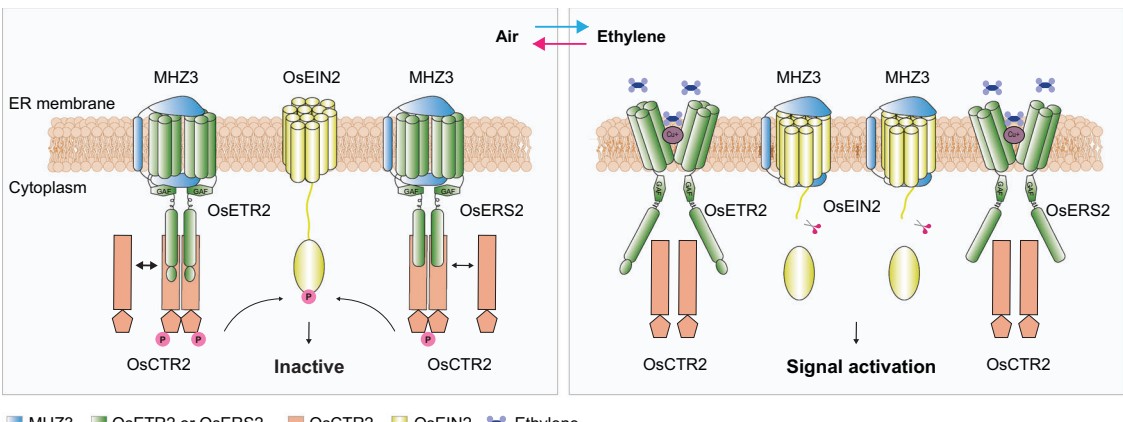

**Fig. 7 | Working model of MHZ3 regulating the on and off states of the ethylene signaling.** Under air conditions, MHZ3 engages with ethylene receptors, enhancing the phosphorylation of OsCTR2, which is tethered to the endoplasmic reticulum membrane via ethylene receptors. Phosphorylated OsEIN2 becomes inactive, turning off ethylene signaling. Upon exposure to ethylene, the interaction between ethylene receptors and MHZ3 weakens, resulting in reduced attachment of OsCTR2 to the endoplasmic reticulum membrane. Consequently, OsCTR2 undergoes dephosphorylation, prompting MHZ3 to transition and stabilize OsEIN2, activating ethylene signaling. After ethylene removal, MHZ3, ethylene receptors, and OsCTR2 reassemble into protein complexes, facilitating the re-phosphorylation of OsCTR2 and deactivating ethylene signaling. In particular, OsETR2 exerted more pronounced regulation over OsCTR2 phosphorylation compared to OsERS2.

## Discussion

Ethylene signaling is initiated upon ligand perception by ethylene receptors, triggering the suppression of the Raf-like kinase AtCTR1. Elucidating the regulatory mechanism of this early signaling event is crucial for understanding the entire signal transduction cascade. In this study, utilizing the rice OsCTR2 phospho-shift as an indicator of signaling activation, we revealed an unexpected role of MHZ3 in ethylene signaling. Apart from acting as a stabilizer of OsEIN2, MHZ3 collaborates with ethylene receptors to regulate the activity of OsCTR2. In air, MHZ3 interacts with the ethylene receptors, enhancing the binding between the ethylene receptors and OsCTR2, resulting in the activation of the ethylene receptor/OsCTR2 complex for OsCTR2 phosphorylation. When exposed to ethylene, the ethylene receptors perceive ethylene and weaken their interactions with MHZ3, leading to a decrease in membrane association of OsCTR2 and a corresponding reduction in its phosphorylation form. Consequently, MHZ3 is switched to stabilize OsEIN2 for ethylene signaling (Fig. 7). The findings not only provide valuable insights into the early events of the ethylene cascade but also may serve as a conceptual paradigm for other signaling pathways, elucidating how a scaffold protein exerts its regulatory function in fine-tuning signaling activation through dynamic interaction with distinct signaling components.

Multiple lines of evidence in this study support the role of MHZ3 in collaboration with ethylene receptors to regulate OsCTR2 phosphorylation, thereby modulating the early response to ethylene signaling. First, the rapid attenuation of ethylene-induced OsCTR2 phosphorylation and its rapid restoration after ethylene withdrawal reflect the initiation and termination of the ethylene signaling cascade, respectively (Fig. 1a, b and Supplementary Fig. 1). Second, MHZ3 physically interacts with OsETR2 and OsERS2 by MbYTH, Co-IP, LCI and domain mapping assays (Fig. 3). Third, genetic analysis suggests that the maintenance of OsCTR2 phosphorylation by MHZ3 and ethylene receptors is mutually dependent (Fig. 4). Fourth, MHZ3 promotes OsETR2's and OsERS2's association with OsCTR2 by semi-in vitro pull-down assays and LCI assays (Fig. 5d–g). Fifth, ethylene was demonstrated to disrupt the association of OsETR2 and OsERS2 with OsCTR2 and MHZ3 in Co-IP assays (Fig. 6a, b), and to reduce the interaction of the two ethylene receptors with MHZ3 in the LCI assays (Fig. 6d–f). Additionally, ethylene decreases the membrane distribution of OsCTR2 mediated by ethylene receptors and MHZ3, as evidenced by the increased presence of non-phosphorylated forms of OsCTR2 (Figs. 1a, b, 2b, 5c, and 6c).

Overall, these pieces of evidence demonstrate that MHZ3 and ethylene receptors collaborate to enhance OsCTR2 phosphorylation, and ethylene perception leads to the dissociation of the MHZ3-receptors-OsCTR2 complex, facilitating the activation of downstream signaling events.

In Arabidopsis, while the interactions between ethylene receptors and AtCTR1 have been demonstrated, and the AtCTR1 kinase activity in relation to EIN2 was studied through in vitro assays[12], how ethylene regulates AtCTR1 activities in vivo was unclear. Our studies found that the membrane protein MHZ3 is absolutely required and sufficient for the OsCTR2 phosphorylation. It should be noted that overexpression of MHZ3 resulted in additional phosphorylation bands of OsCTR2, supporting that MHZ3 promotes OsCTR2 phosphorylation possibly at multiple sites. Additionally, treatment with 1-MCP alone, which inhibits the perception of endogenous ethylene, also enhances the phosphorylation of OsCTR2. Considering the rapid change of OsCTR2 phosphorylation status, we hypothesize the presence of phosphatases responsible for the dephosphorylation of OsCTR2 to fine-tune its activity. Further research may explore whether such phosphatases exist and how they modulate OsCTR2 dephosphorylation.

Investigation of ethylene receptor loss-of-function single mutants suggests that OsETR2 may outperform other ethylene receptors, e.g. OsERS2, in maintaining OsCTR2 phosphorylation (Fig. 2a). In Arabidopsis, the histidine kinase domain of AtETR1 alone can interact with AtCTR1, while the addition of the receiver domain strengthened this interaction. Additionally, the affinity between AtETR1 containing the receiver domain and AtCTR1 is stronger than that of AtERS ethylene receptors lacking the receiver domain[46]. Presently, in addition to containing the receiver domain, OsETR2 also possesses serine/threonine protein kinase activity[28]. The structural and biochemical characteristics distinctions between OsETR2 and other ethylene receptors may result in varied regulation of OsCTR2 phosphorylation. Rice contains three OsCTRs, namely OsCTR1, OsCTR2, and OsCTR3[29]. It is likely that different ethylene receptors may exhibit varying affinities for these OsCTRs proteins, necessitating further research.

Our previous study discovered that the phosphorylation status of OsCTR2 remains fixed in the gain-of-function mutant $Osers2^d$[30]. The mutation site of $Osers2^d$ is located within the ethylene-binding region of the transmembrane domain, suggesting the importance of ethylene recognition for inactivation or conformational change of ethylene receptors. Ethylene receptors exist as clusters in plants[47], whether the dominant active version of $Osers2^d$ can cooperate with other ethylene

receptors, such as OsETR2, to maintain a fixed phosphorylation state of OsCTR2 requires further investigation.

It should be mentioned that, although MHZ3 interacts with ethylene receptors to promote OsCTR2 phosphorylation, how MHZ3 affects ethylene receptor signaling output and thus impacts the phosphorylation of OsCTR2 is still unclear. It is possible that MHZ3 helps maintain the active conformation of the ethylene receptors, thereby stabilizing ethylene receptor-OsCTR2 complex until ethylene exposure alters this binding equilibrium. Furthermore, it remains to be determined whether the serine/threonine kinase activity and/or histidine kinase activity inherent in ethylene receptors is regulated by MHZ3, thereby affecting the signal output of these receptors. Given that MHZ3 mainly interacts with the transmembrane region and GAF domain of ethylene receptors, it remains plausible that MHZ3 directly affects the receptor's binding ability to ethylene and 1-MCP. In Arabidopsis, AtRAN1 plays a crucial role in ethylene receptor biogenesis and binding activity by regulating copper delivery[48]. Whether MHZ3 plays a similar role requires further investigation. Besides MHZ3, a series of ethylene receptors-interacting proteins have been identified from different species, such as AtRTE1 and AtARGOS in Arabidopsis[49–51], and NtNEIP2 and NtTCTP in tobacco[52,53]. Investigating their relationship with MHZ3 in regulating ethylene receptor function will shed light on the mechanisms involved.

With ethylene acting as a plant hormone for over 450 million years and MHZ3 orthologs conserved from algae through land plants[3,33], MHZ3 likely constitutes an ancient regulator of ethylene signaling. Whether the Arabidopsis homolog also affects AtCTR1 phosphorylation requires further investigation. An intriguing evolutionary question is whether the dual functions of MHZ3 as an ethylene receptor interactor and EIN2 stabilizer are closely coupled, or can be separated in different plant species.

Our previous studies have found that MHZ3 mutations result in complete ethylene insensitivity in both coleoptile and root growth[33]. Currently, we find that the MHZ3 mutation leads to a loss of OsCTR2 phosphorylation, which is similar to the phosphorylation state of OsCTR2 in the ethylene receptor loss-of-function single mutant *Osetr2*, and double mutants *Osetr2 ers2* and *Osetr2 etr3* (Fig. 2a). From these, it seems that the loss of OsCTR2 phosphorylation, which should facilitate ethylene signaling and response, is not consistent with the ethylene insensitivity in *mhz3* mutants. This may be because OsCTR2 acts upstream of OsEIN2. Mutation of MHZ3 results in ubiquitination-mediated degradation of OsEIN2, which is epistatic to OsCTR2[33]. It is interesting to find that, compared to the WT, the length of coleoptiles in *mhz3-1* treated with the ethylene perception inhibitor 1-MCP for three days was not inhibited but significantly longer than the wild type under the same treatment (Supplementary Fig. 4a). This phenomenon most likely reflects the residual constitutive ethylene response phenotype after losing the OsCTR2 phosphorylation activity in *mhz3*. Further supporting MHZ3's role beyond OsEIN2 stabilization, overexpressing OsEIN2-GFP in the *mhz3* mutant led to a constitutively ethylene-responsive phenotype. However, this overexpression did not rescue the *mhz3* mutant's insensitivity to exogenous ethylene or 1-MCP treatment (Supplementary Fig. 17). These results strongly indicate that MHZ3 is involved in regulating ethylene perception mediated by ethylene receptor complexes.

Our previous results showed that despite a significant accumulation of OsEIN2 protein in MHZ3-overexpressing (MHZ3-OE) lines, these lines only exhibited a mild ethylene hypersensitive response phenotype[33]. In our study, we observed enhanced OsCTR2 phosphorylation in MHZ3-OE lines, potentially promoting the phosphorylation modification of OsEIN2 and its subsequent inactivation. The antagonistic regulation of MHZ3 on OsEIN2 protein stability and activity may be the reason behind the mild phenotype observed in MHZ3-OE. This also provides another genetic evidence for MHZ3's involvement in regulating OsCTR2 phosphorylation.

While EIN2 plays a central role in ethylene signaling, how it is regulated by upstream signals is not well understood. Specifically, while AtCTR1-mediated phosphorylation of EIN2 and ubiquitination-mediated degradation of EIN2 are often discussed together in current models[4], whether these two processes are causally linked remains largely unclear. Our data show that in the *mhz3* mutant, where OsCTR2 kinase activity is largely suppressed, OsEIN2 protein is highly unstable and undergoes ubiquitin-mediated degradation[33]. This result suggests that phosphorylation of OsEIN2 is not the direct cause of its degradation. Therefore, more evidence is required to determine the relationship between EIN2 phosphorylation and its degradation.

MHZ3 plays dual roles in ethylene signal transduction, exhibiting both positive and negative effects. It is involved in stabilizing the OsEIN2 protein[33] and collaborates with ethylene receptors to regulate the phosphorylation of OsCTR2. It is not uncommon that a single protein can integrate complex functions, with some playing opposing roles in the same signaling pathway[41,54–57]. Interestingly, Arabidopsis AtCTR1 also plays dual roles in ethylene signaling. It has a negative regulatory role on the endoplasmic reticulum membrane, but it can exert a positive regulatory function when entering the nucleus[41]. Traditional genetic screen-based studies are susceptible to overlooking the multifunctionality of proteins due to their reliance on phenotype-based screening systems. For instance, the impact of MHZ3 on the ethylene receptor-CTRs complex may be over-shadowed by its influence on OsEIN2. In our study, the utilization of OsCTR2 phospho-shift as a molecular output for ethylene signaling activation provided an ideal approach for identifying and studying the dual function of MHZ3.

Collectively, we find that MHZ3 associates with ethylene receptors to promote OsCTR2 phosphorylation, and identifies a previously unknown mechanism for early signaling in the protein complex involving MHZ3, ethylene receptors, and OsCTR2.

## Methods
### Plant materials and growth conditions
The rice (*Oryza sativa* L.) materials *mhz3*, *mhz7-1/Osein2-1*, *mhz6/Oseil1*, *mhz11*, *mhz12/Osers2^d* (a dominant gain-of-function variant of OsERS2[A32V], analogous to *Arabidopsis thaliana Atetr1-3*)[27,58], *mhz3-1/gMHZ3* (a *MHZ3*-complementation line), *mhz3/MHZ3-GFP* (Overexpression of MHZ3-GFP in *mhz3*), MHZ3-OE (MHZ3 over-expression transgenic lines), OsEIN2-GFP, and OsEIN2-GFP/*mhz3* were previously created in our laboratory[30,31,33]. T-DNA insertion knockout mutants of *Osers1*, *Osers2*, *Osetr2*, and *Osetr3* were purchased from the POSTECH Biotech Center and identified in our previous study[59]. Double homozygous materials of *Osetr2 mhz3*, *Osers2 mhz3*, *Osetr2 ers2*, *Osetr2 etr3*, *Osers2^d mhz3-1*, and MHZ3-GFP/*Osetr2* were generated by crossing. The rice plants utilized in this study were cultivated at the Experimental Station of the Institute of Genetics and Developmental Biology in Beijing from May to October, followed by cultivation in the Hainan Experimental Station from November to the subsequent April each year. Ethylene treatment involved placing uniformly germinated rice seeds, pretreated at 37 °C, on stainless steel sieves within 5.5-L airtight plastic containers supplied with varying concentrations of ethylene. The seedlings were then cultivated in darkness at 28 °C for 2-3 days to evaluate the growth of their coleoptiles and roots[31]. Rice seedlings designated for protein extraction were grown in darkness for 2 days and subsequently treated for specified durations according to the experimental conditions. For instance, treatments with 10 μL/L ethylene and 10 μL/L 1-MCP lasted for 1 hour before samples were collected for OsCTR2 phosphorylation analysis. Etiolated rice seedlings used for protoplast isolation were cultivated in 1/2 MS medium (1/2 MS salt, 1% sucrose, and 0.3% phytagel) under dark conditions at a temperature of 28 °C for a growth period of 7-10 days.

## Epistasis analysis

*Osetr2* mutant is in the Dongjin (DJ) background, and *mhz3-1, Osers2[d], and* MHZ3-GFP are in the Nipponbare (WT) background. *Osers2[d] mhz3-1* and MHZ3-GFP/*Osetr2* were generated by crossing. *35 S:OsETR2-Myc* and *35 S:OsERS2-Myc* were transgenic rice plants in WT (Nipponbare, OsETR2-Myc/WT and OsERS2-Myc/WT) and *mhz3* (OsETR2-Myc/*mhz3* and OsERS2-Myc/*mhz3*) backgrounds, respectively. Epistatic relationships were judged by examining the phosphorylation state of OsCTR2 in these plants.

## Gene expression analysis

Etiolated seedlings were harvested to extract total RNA using TRIZOL reagent (Thermo Fisher Scientific, Trizol). The cDNAs were synthesized utilizing the Maxima H Minus cDNA Synthesis Master Mix (Thermo Fisher Scientific, M1682), and qRT-PCR analyzes were performed employing TB Green® Premix Ex Taq™ (Tli RNaseH Plus) (TaKaRa, RR420D), with *OsUBQ5* serving as an internal control for normalization. Each data point at least has three biological replicates.

## Antibody generation and immunoblot analysis

Polyclonal antibody specific to OsCTR2 was produced by immunizing mice with synthetic peptides conjugated to keyhole limpet hemocyanin (KLH). The peptide sequence DKGGDPADRPAGSSGGGG from OsCTR2's N-terminus was used for immunization. To produce the MHZ3 antibody, the cDNA fragment encoding amino acids 21 to 270 of MHZ3 was inserted into the pQE30 Xa vector (Qiagen) and expressed in *E. coli* M15. Subsequently, the recombinant protein was purified using a HisTrap-HP column (Amerhsam) under denaturing conditions. All the above antibodies were produced at the Experimental Animal Center of the Institute of Genetics and Developmental Biology, CAS.

For the immunoblot analysis, the proteins were denatured at 65 °C for 5 min in SDS-PAGE loading buffer and then separated using SDS-PAGE. They were subsequently transferred to a polyvinylidene difluoride membrane (PVDF, IPVH00010, Merck Millipore) using the semi-dry transfer membrane method at 25 V for 60 min. Following the transfer, the membrane was blocked for 2 h at room temperature with PBS containing 5% skim milk powder. After blocking, the membrane was incubated with the primary antibody. The primary antibodies used include: anti-OsCTR2 (1:10,000), anti-MHZ3 (1:10,000), anti-GFP (7G9) (1:10,000; M2004, Abmart), anti-BiP (1:20,000; ER marker; AS09 481, Agrisera), anti-UGPase (1:20,000; cytoplasm marker; AS05 086, Agrisera), anti-histone H3 (1:20,000; nuclear marker; AS10710, Agrisera), anti-c-Myc (19C2) (1:10,000; HRP-conjugated, M20019L; Abmart), anti-FLAG (3B9) (1:5000; M20008, Abmart), and anti-Luciferase (clone LUC-1, 1:5,000, L2164-.2 ML, Sigma-Aldrich). The anti-OsCTR2 was diluted in Immunoreaction Enhancer Solution I (NKB-201, Toyobo), while other primary antibodies were diluted in 3% (w/v) skim milk powder dissolved in PBS. Secondary goat anti-rabbit or anti-mouse-IgG-horseradish peroxidase (M210011, M210021, Abmart) antibodies were used at 1:10,000 dilutions in PBS containing 3% (w/v) skim milk powder for 1 h at room temperature. The signals were detected by chemiluminescence method using WesternBright ECL Detection Kit (K-12045-D50; Advansta). When needed, the signal intensities were quantified using ImageJ software with default parameters (National Institutes of Health). The 4-15% precast polyacrylamide gel (4561083, 4561085, Biorad) was employed to enhance the separation efficiency between OsCTR2-P and OsCTR2.

## Subcellular colocalization analysis

For analyzes of the co-localization of MHZ3 and ethylene receptors, we transiently expressed 35 S: *MHZ3-GFP* and 35 S: *OsETR2-mCherry*, as well as 35 S: *MHZ3-GFP* and 35 S: *OsERS2-mCherry*, in *Nicotiana benthamiana* leaf epidermal cells through Agrobacterium infection and in rice protoplasts. The images were taken using confocal microscopy (Zeiss TIRF3). Excitation/emission wavelengths were set at 488 nm/500–530 nm for GFP and 561 nm/582-639 nm for mCherry.

## Split-ubiquitin membrane yeast two-hybrid system (MbYTH)

The principle of MbYTH is based on Stagljar et al.[42] *MHZ3*-coding sequence was cloned into the bait vector pBT3-SUC (MHZ3-Cub), and *OsETR2* and *OsERS2*-coding sequence was cloned into the prey vector pPR3-N (NubG-OsETR2 and NubG-OsERS2) from the DUAL membrane starter kit SUC (Dualsystem Biotech). Yeast strain NMY51 cells were cotransfected with the bait and prey constructs. To test for self-activation, the empty bait and prey vectors were cotransfected with the MHZ3-Cub, NubG-OsETR2, and NubG-OsERS2 constructs, respectively. The wild-type N-terminal half of ubiquitin NubI (pOst1-NubI) was cotransfected with MHZ3-Cub to detect the functional expression of MHZ3 protein. The positive transformants were screened on the SD-Trp-Leu medium, and protein interactions were tested on the SD-Trp-Leu-His medium. The growth of yeast colonies was evaluated after 5 days of incubation at 30 °C.

## Co-immunoprecipitation (Co-IP) assay

Co-IP assays in this study were based on Ma et al.[33] In the transient expression system using protoplasts, the constructs in various combinations were co-transformed into the protoplasts of rice seedlings for Co-IP analysis. The protoplasts were incubated at 28 °C for 16 hours in the dark, and total protein was extracted with IP buffer [50 mM HEPES (pH 7.5), 150 mM NaCl, 0.5 mM EDTA, 0.5% NP-40, 50 μM MG132, 2% (v/v) protease inhibitor cocktail (Sigma)]. Incubate the samples on ice for 20 min, vortexing every 5 min. The 10% supernatant obtained by centrifuging twice at 16,000 *g* for 10 min at 4 °C was used as input and the remaining supernatant was incubated with 30 μL equilibrated GFP-Trap_A (gta-20, Chromotek) beads at 4 °C for 1 h. Subsequently, the beads were washed 5 times at 4 °C with washing buffer [100 mM KCl, 2.5 mM MgCl2, 10 mM HEPES (pH 7.4), 10% glycerin, 5 ‰ (v/830 v) NP40, 50 μM MG132, 2 mM PMSF, 2% (v/v) protease inhibitor cocktail (Sigma)]. The eluted immunoprecipitates were immunoblotted with anti-GFP, anti-c-Myc, anti-FLAG, and anti-BiP antibodies.

For Co-IP assay in stable transgenic plants, different plant backgrounds and treatment conditions were selected according to experimental requirements. Total proteins were immunoprecipitated with anti-c-Myc affinity gel (E6654, Sigma-Aldrich) or with GFP-Trap_A beads. The subsequent steps are the same as described above.

## Semi-in vitro pull-down (PD) assay

OsETR2-Myc and OsERS2-Myc proteins were immunoprecipitated with anti-c-Myc affinity gel from OsETR2-Myc/*mhz3* and OsERS2-Myc/*mhz3* stable transgenic plants, respectively. WT, *mhz3*, and MHZ3-OE#22 etiolated seedlings were grown in the dark for 2 days followed by freezing in liquid nitrogen. Subsequently, the frozen samples were finely powdered and weighed equally. Total protein extraction was performed using IP buffer as described above, with the resulting supernatant being retained as homogenate. Equal amounts of OsETR2-Myc and OsERS2-Myc proteins were introduced into the homogenates and incubated at 4 °C for 1 hour. Subsequent procedures were the same as Co-IP assays.

## Total and membrane protein isolation

Total and membrane proteins were isolated according to Ma et al.[33] Briefly, 3 grams of seedlings were ground into a fine powder using liquid nitrogen and subsequently dissolved in 5 mL of extraction buffer [100 mM of Tris-HCl (pH 8.0), 150 mM of KCl, 5 mM of EDTA, 10% Glycerol (v/v), 3.3 mM dithiothreitol (DTT), 0.6% (w/v) polyvinylpyrrolidone (PVPP), and 2% (v/v) protease inhibitor cocktail (Sigma-Aldrich)]. Homogenates were filtered through two layers of

Miracloth (475855-1 R, Merck Millipore) and then centrifuged at 16,000 g for 10 min to eliminate debris. Supernatants were further centrifuged at 100,000 g for 60 min at 4 °C and the microsomal membrane was subsequently washed three times with the extraction buffer and then dissolved in 100 μL of extraction buffer, which was augmented with 1% (v/v) Triton X-100 and 0.1% (w/v) SDS. This was achieved by incubating the mixture on ice for 20 min. Samples were heated at 65 °C for 5 min with SDS-PAGE loading buffer before loaded for SDS-PAGE and immunoblot analysis.

## Nuclear protein isolation

The nuclei protein isolation was conducted using the CelLytic PN Plant Isolation/Extraction Kit (Sigma-Aldrich, CELLYTPN1-1KT). Five grams of shoot tissue from 2-day-old etiolated rice seedlings were weighed and ground in liquid nitrogen. The ground tissue was then mixed with 15 mL of nuclear isolation buffer (NIB) and filtered through a 100-mesh sieve along with miracloth. The lysate (total protein) was centrifuged at 1260 g for 20 min to separate the cytoplasmic and nuclear fractions. NIBA containing 0.3% Triton X-100 was used to lyse cell membranes, then washed the crude nuclei with NIBA 8 times. A semi-pure nuclei preparation was obtained using a 2.3 M sucrose cushion. Following this, nuclear proteins were extracted from the suspended nuclei using NIBA-diluted 1x SDS-PAGE loading buffer.

## Luciferase complementation imaging (LCI) assay

LCI assay is used to study protein-protein interactions[43]. The Cluc-OsETR2, Cluc-OsERS2, OsCTR2-Nluc, MHZ3-Nluc, GFP-Nluc, GFP, and MHZ3-GFP constructs were transformed into Nicotiana benthamiana leaf cells through A. tumefaciens strain GV3101. Transformed Nicotiana benthamiana plants were grown in a greenhouse at 22 °C under a 15-h light/9-h dark photoperiod. LCI assay images were captured using a low-light, cooled charge-coupled device imaging apparatus (Night OWL II LB983 with indiGO software; Berthold, Germany). For the enhanced-interaction assay, Agrobacterium carrying OsETR2 or OsERS2 with the Cluc tag was mixed with Agrobacterium carrying OsCTR2 with the Nluc tag. Subsequently, the mixture was co-infiltrated with the control vector GFP or MHZ3-GFP on both sides of the same Nicotiana benthamiana leaf. The protein expression levels of Cluc-OsETR2 and Cluc-OsERS2 were detected by anti-Luciferase antibody. Also, the protein expression levels of OsCTR2-Nluc and MHZ3-Nluc were detected by anti-OsCTR2 antibody and anti-MHZ3 antibody, respectively.

## Statistical analysis

Statistical significance was assessed using a two-tailed Student's t-test. Data are presented as means ± SD, with *$P < 0.05$ indicating a P-value of less than 0.05, and *$P < 0.01$ indicating a P-value of less than 0.01. Detailed analysis results can be found in the Source data file.

## Reporting summary

Further information on research design is available in the Nature Portfolio Reporting Summary linked to this article.

# Data availability

All study data are included in the article and/or supporting information. Source data are provided with this paper.

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

## Acknowledgements

This work is supported by the STI 2030-Major Projects (2023ZD0407102) and the National Natural Science Foundation of China (32370333, 31600980, and 3220025).

## Author contributions

X.-K.L., J.-S.Z., H.Z., C.-C.Y. and S.-Y.C. designed the research; X.-K.L. performed most of the research; B.M. identified *MHZ3* gene and performed Fig. 3c (right); all authors including X.-K.L., J.-S.Z., Z.H., C.-C.Y., S.-Y.C., Y.-H.H., R.Z., W.-Q.C., L.L., J.-Q.H., Y.Z., X.Z., W.-A.W., J.-J.T., W.W., W.-K.Z., and B. M. contributed to material preparation, data analysis and discussion; X.-K.L., H.Z, and J.-S.Z. wrote the article.

## Competing interests

The authors declare no competing interests.
