## [Peer Review File · Nature Communications]

Reviewers' Comments:

Reviewer #1:

Remarks to the Author:

MHZ3, previously identified as a stabilizer of OsEIN2, is crucial to maintain OsCTR2 phosphorylation in absence of ethylene. Ethylene exposure triggers phosphoreduction of CTR2, a shift to its non-phosphorylated form.

The mechanism of this shift was previously unknown. MHZ3 interacts with both subfamily I and II ethylene receptors in rice (OsERS2 and OsETR2). The key point of the work is the demonstration of the dual role of MHZ3, as an on-off switch: off, by stabilizing the receptor-OsCTR2 complex through interaction with the receptors (MHZ3 is essential for the binding of receptors OsETR2 and ERS2 with phosphorylated OsCTR2), blocking signaling, and on, upon ethylene exposure, by interaction with and stabilization of EIN2, enabling signaling. In the absence of the hormone, MHZ3 associates with the receptors to facilitate their interaction with OsCTR2 (hence also temporarily associating CTR2 to the ER membrane), while in the presence of the hormone, the interaction of MHZ3 with the receptors is attenuated, resulting in a suppression of the activity of the negative regulator CTR2, with MHZ3 associating to EIN2, triggering downstream response. Hence, ethylene alters the binding equilibrium, possibly by suppression/induction of the binding partners. MHZ3 appears necessary and sufficient for OsCTR2 phosphorylation.

This work provides novel insight into the early events in ethylene signaling, covering alterations in spatiotemporal dynamics of signaling factors, which is not only important for the ethylene field, but since MHZ3 plays a dual role, also serves as a conceptual paradigm for other signaling pathways. It might also be important in further understanding of interactions with other plant hormone signaling pathways. From the point of view of spatiotemporal dynamics, it extends the findings published in NComm last year on subcellular trafficking of CTR1, supporting a dual role of AtCTR1, which was previously only seen as a negative regulator of ethylene signaling. Therefore, it is also of interest to a broader readership. The study is generally well-conducted. Most of the evidence is based on protein-protein interactions, which are supported by Y2H, Co-IP, Luc-complementation imaging, and domain mapping assays.

While the findings are novel and deserve publication, there are a few important elements to be considered. In addition, the manuscript is not ready to be easily crunched by non-specialist readers.

Major points:

1. RE: effect of 1-MCP on the phosphorylation state of CTR2: it is not fully clear whether the results in Figure 1C and others experiments where 1-MCP was used show the action of 1-MCP only or in combination with ethylene. Also, in order to confirm whether observed effects reflect an inhibitor's function, a complementation experiment should be done by combining ligand and inhibitor. The ratio between the phosphorylated and non-phosphorylated forms is different as compared to the control, with more of the phosphorylated form accumulating in 1-MCP treated plants (maintenance of the phosphorylated state). Since 1-MCP blocks receptor action, receptor function or a conformational change in the receptor following ethylene binding, is necessary to trigger dephosphorylation of CTR2. Maintenance of phosphorylation state is also seen in the *ers2-D* mutant, this should be addressed in the discussion, offering possible interpretations, and possibly linking this to the importance of the sensor versus receiver domains in receptors.
2. In the discussion, the particular role of OsETR2 should be more explicit. A very interesting point is the differentiability of OsCTR2 phosphorylation in *ers1* and *ers2* mutants compared to *etr2*. It opens several routes of interpretation, which should be discussed in more detail, and reflected in the final scheme in figure 6.
3. Regarding the presentation of immunoblot data. It is not best practice to show a restricted part of a blot; zoom into the bands of interest in the core manuscript but provide complete blots as supplemental material. Background signals are not necessarily a problem and can even be a reference in certain cases. Size markers also provide a good reference point. For instance, it is not

clear in Fig 1i whether the indicated bands are corresponding to the non-phosphorylated form of CTR2. On the other hand, there appears to be a third band, on top of CTR2-P, and sometimes, more of the higher MW bands appear. This is also seen in the overexpressor lines and complemented loss of function mutant, as well as in 1-MCP treated WT in Suppl 3b. This is probably not coincidental; to what can these correspond?

4. RE: Accessibility to non-specialist readers: in order to allow direct interpretation of mutant phenotypes, not only the results section but also legends should systematically mention the nature of mutations (loss/gain), in the current version it is sometimes clarified but not consistently. Legends should be improved and all need to be revised to have every acronym mentioned in the figure in full. In addition, the concentration of ethylene, as well as of 1-MCP, has to be mentioned in every figure legend, as well as in the method description. Legend to Figure 3: Y2H for membrane based proteins: legend lacks information for the reader not familiar with the split ubiquitin systems. The authors also failed to refer to the original paper by Stagljar in PNAS in the manuscript, which has to be included. Again, all acronyms used in the figure panels are to be included in full in the legend. LCI assays are used in several experiments; mention Luciferase complementation assays in full in legends. Still on the legend to Figure 3. For non-specialist readers it is also more informative to indicate the proper domain structure of ethylene receptors with Ethylene binding -GAF- His-kinase – Receiver domain. The legend does not mention what REC is, and His-KA is not a standard acronym for Histidine kinase, rather HK. Clarify in legend and again, acronyms are to be mentioned in full.

Minor:

1. Line 35: the ethylene receptor/CTR1 complex
2. Line 105: the sentence should be moved up
3. Line 360: refers to AtCTR1, rather than using CTR1 without reference to the species. The latter can be very confusing since OsCTR1 exists too, yet OsCTR2 is the closest homolog of AtCTR1, not OsCTR1. A general check of the manuscript on this should be performed.
4. Line 416: this statement linking the predominant role of ETR2 in regulating OsCTR2 phosphorylation to the presence of a receiver domain, seems to lack a reference, since this manuscript does not present evidence for the binding of ETR1 neither EIN4.
5. Supplemental figures 12 and 13 contain information that deserves to be part of the core manuscript. Authors could combine S12 for instance with Fig1. Fig.S13 could be part of Fig.6.
6. Figure 3: 'Prey' is with an 'e' in this context, not 'a'.
7. The origin of all antibodies used in immunoblots should be clearly mentioned in the Materials section, including company or reference from lab of origin.

Reviewer #2:

Remarks to the Author:

In this paper, the authors reveal that MHZ3 is indispensable for the ethylene receptor-mediated phosphorylation of OsCTR2 in rice, which seems important for ethylene signaling initiation. The authors found that MHZ3 could interact with ethylene receptors and facilitate the interactions between receptors and OsCTR2, which were inhibited by ethylene treatment. Uncovering the mechanistic role of MHZ3 in ethylene signaling is valuable since *mhz3* mutants are completely insensitive to ethylene both in coleoptile and root. Overall, the results presented in this manuscript are well-organized and convincing. However, there are some key questions mainly regard to the biological significance of this regulation remained to be addressed.

Major concerns:

1. Zhao et al., (Plant cell, 2020) had already reported the existence of OsCTR2 phosphorylation and ethylene treatment decreased the phosphorylation level of OsCTR2. In this paper, the author

further confirmed the rapid phosphorylation changes of OsCTR2 upon the switch on/off of ethylene signaling cascade. It's very likely that the phosphorylation status of OsCTR2 may affect its kinase activity and thus regulate the downstream ethylene signaling. However, the authors only demonstrated a correlation between OsCTR2 phosphorylation and ethylene response, there are no direct results supporting the importance of OsCTR2 phosphorylation. The authors should identify the phosphorylation sites of OsCTR2 and construct related genetic materials to address the biological relevance of OsCTR2 phosphorylation in ethylene signaling.

2. It seems that the non-phosphorylated OsCTR2 is inactive and could release the repression of downstream ethylene signaling through OsEIN2. The authors found that OsCTR2 phosphorylation is completely abolished by mhz3 mutation in the air and couldn't respond to ethylene or 1-MCP treatment as well. Theoretically, ethylene signaling should be constitutively activated in mhz3 mutants. However, the ethylene response is deprived in mhz3.

The authors claimed that mutation of MHZ3 could also result in the ubiquitination-mediated degradation of OsEIN2 as reported by Ma et al., 2018, which is epistatic to OsCTR2. It's reasonable, however, weaken the significance of the regulation of MHZ3 on OsCTR2. The authors provide strong evidences demonstrating that MHZ3 is indispensable for OsCTR2 phosphorylation, but, likewise, the biological relevance of this regulation is unclear. The authors should try to distinguish the two opposite roles of MHZ3 genetically.

Minor concerns:

1. Figure 3a, the resolution is not enough to distinctly show the colocalization of OsMHZ3 and ethylene receptors. The authors should provide higher resolution images.
2. Figure 3c and figure 5d, there are additional GFP bands in the experimental materials expressed MHZ3-GFP, sometimes in input, sometimes in the IP product, why?
3. Figure 3f, the authors should indicate which band represents the OsERS2-GAFHK in the input group.
4. Figure 4b upper panel, the quality is too low to distinguish the phosphorylated and non-phosphorylated CTR1 especially in WT. The authors should replace this figure with better quality images.
5. Figure 4c and d, the phosphorylation status of CTR1 in WT should be added to indicate the promotion effects of ethylene receptors overexpression.
6. Line 171, identity. (Fig. 1i), "." should be deleted.
7. Line 190, "Under" should be "under"

Responses to the REVIEWERS COMMENTS

For Reviewer #1 (Remarks to the Author):

MHZ3, previously identified as a stabilizer of OsEIN2, is crucial to maintain OsCTR2 phosphorylation in absence of ethylene. Ethylene exposure triggers phosphoreduction of CTR2, a shift to its non-phosphorylated form.

The mechanism of this shift was previously unknown. MHZ3 interacts with both subfamily I and II ethylene receptors in rice (OsERS2 and OsETR2). The key point of the work is the demonstration of the dual role of MHZ3, as an on-off switch: off, by stabilizing the receptor-OsCTR2 complex through interaction with the receptors (MHZ3 is essential for the binding of receptors OsETR2 and ERS2 with phosphorylated OsCTR2), blocking signaling, and on, upon ethylene exposure, by interaction with and stabilization of EIN2, enabling signaling. In the absence of the hormone, MHZ3 associates with the receptors to facilitate their interaction with OsCTR2 (hence also temporarily associating CTR2 to the ER membrane), while in the presence of the hormone, the interaction of MHZ3 with the receptors is attenuated, resulting in a suppression of the activity of the negative regulator CTR2, with MHZ3 associating to EIN2, triggering downstream response. Hence, ethylene alters the binding equilibrium, possibly by suppression/induction of the binding partners. MHZ3 appears necessary and sufficient for OsCTR2 phosphorylation.

This work provides novel insight into the early events in ethylene signaling, covering alterations in spatiotemporal dynamics of signaling factors, which is not only important for the ethylene field, but since MHZ3 plays a dual role, also serves as a conceptual paradigm for other signaling pathways. It might also be important in further understanding of interactions with other plant hormone signaling pathways. From the point of view of spatiotemporal dynamics, it extends the findings published in NComm last year on subcellular trafficking of CTR1, supporting a dual role of AtCTR1, which was previously only seen as a negative regulator of ethylene signaling. Therefore, it is also of interest to a broader readership. The study is generally well-conducted. Most of the evidence is based on protein-protein interactions, which are supported by Y2H, Co-IP, Luc-complementation imaging, and domain mapping assays.

While the findings are novel and deserve publication, there are a few important

elements to be considered. In addition, the manuscript is not ready to be easily crunched by non-specialist readers.

【Response】 Thank you very much for the valuable comments and encouragement. We have revised the MS with more experiments according to the suggestions (Please see the following responses). We also improved the readability of our article by labeling the molecular weights of corresponding proteins, minimizing the use of abbreviations but opting for full spellings in the figure legends. Specific treatment concentrations and durations are indicated in the figure captions. Additionally, we have tried to use simple and common words throughout the manuscript. Some of the comments were also incorporated into the discussion parts at lines 391-395.

Major points:

1. RE: effect of 1-MCP on the phosphorylation state of CTR2: it is not fully clear whether the results in Figure 1C and others experiments where 1-MCP was used show the action of 1-MCP only or in combination with ethylene. Also, in order to confirm whether observed effects reflect an inhibitor's function, a complementation experiment should be done by combining ligand and inhibitor.

【Response】 Thank you for the comment. In Fig. 1c and other results, the name of '1-MCP' alone indicates its individual use. We have provided corresponding explanations in the respective figure legends.

According to your valuable suggestions, we presented complementary experimental results of the combined use of ethylene and the ethylene receptor inhibitor, 1-MCP. The combined use caused a partial reduction in ethylene-induced phosphorylation of OsCTR2, leading to a partial suppression of the induction of corresponding ethylene-responsive genes (Supplementary Fig. 5). The results have been incorporated into the manuscript, appearing on page 6, lines 154-158.

The ratio between the phosphorylated and non-phosphorylated forms is different as compared to the control, with more of the phosphorylated form accumulating in 1-MCP treated plants (maintenance of the phosphorylated state). Since 1-MCP blocks receptor action, receptor function or a conformational change in the receptor following ethylene

binding, is necessary to trigger dephosphorylation of CTR2. Maintenance of phosphorylation state is also seen in the *ers2-D* mutant, this should be addressed in the discussion, offering possible interpretations, and possibly linking this to the importance of the sensor versus receiver domains in receptors.

【Response】 Thank you for your very good suggestions. We have added relevant descriptions in the discussion section (Lines 424-426) to help better understand the functional mechanism at the very initial signaling stage of ethylene-receptor-CTR2. We have also added a discussion on the potential mechanism maintaining OsCTR2 phosphorylation in *Osers2^d* (Lines 446-453) according to your inspiration.

2. In the discussion, the particular role of OsETR2 should be more explicit. A very interesting point is the differentiability of OsCTR2 phosphorylation in *ers1* and *ers2* mutants compared to *etr2*. It opens several routes of interpretation, which should be discussed in more detail, and reflected in the final scheme in figure 6.

【Response】 Thank you very much for your advice. We discussed the larger role of OsETR2 in regulating OsCTR2 phosphorylation in terms of its structural and biochemical properties in the discussion section (Lines 432-444). Based on your suggestion, we have modified the final scheme (Fig.7).

3. Regarding the presentation of immunoblot data. It is not best practice to show a restricted part of a blot; zoom into the bands of interest in the core manuscript but provide complete blots as supplemental material. Background signals are not necessarily a problem and can even be a reference in certain cases. Size markers also provide a good reference point.

【Response】 Thank you very much for your advice. We have provided more detailed molecular weight markers for all proteins mentioned throughout the manuscript. Additionally, all original, uncropped blot and statistical data from this study are presented in the Source Data file.

For instance, it is not clear in Fig 1i whether the indicated bands are corresponding to

the non-phosphorylated form of CTR2.

【Response】 Thank you very much for your advice. To address this issue, we have added Fig. 1i, which shows that λ -PPase treatment can eliminate phosphorylation of OsCTR2 in the wild-type. The dephosphorylated OsCTR2 matches non-phosphorylated OsCTR2, as explained in lines 181-182 of the manuscript. Additionally, since OsCTR2 phosphorylation is absent in the *mhz3* mutant, the molecular weight of OsCTR2 after λ -PPase treatment in the wild-type and MHZ3-OE aligns with that of OsCTR2 in *mhz3*, indicating that the bands in the lower panel of Fig. 1j (initially manuscript Fig. 1i) denote the non-phosphorylated form of OsCTR2.

On the other hand, there appears to be a third band, on top of CTR2-P, and sometimes, more of the higher MW bands appear. This is also seen in the overexpressor lines and complemented loss of function mutant, as well as in 1-MCP treated WT in Suppl 3b. This is probably not coincidental; to what can these correspond?

【Response】 Thank you very much for the comments. In Fig. 1j, we observed higher molecular weight bands of OsCTR2 in the overexpression lines of *mhz3*/MHZ3-GFP and MHZ3-OE, which weaken upon ethylene treatment. Additionally, treatment with λ -PPase eliminates these additional modifications. These results suggest that the additional bands of OsCTR2 in the MHZ3 overexpression lines are phosphorylation modifications. We label these as OsCTR2-P^{super} in Fig. 1j and Fig. 4b, as explained in lines 181-182 of the manuscript. We also noted in Supplementary Fig. 5 that treatment with 1-MCP alone increases the ratio of OsCTR2-P/OsCTR2 and suppresses the activation of downstream genes. This may be because 1-MCP can inhibit the binding of ethylene to receptors, leading to phosphorylation modifications at more sites on OsCTR2. In the discussion section, we also elaborate on this phenomenon from lines 424 to 426.

4. RE: Accessibility to non-specialist readers: in order to allow direct interpretation of mutant phenotypes, not only the results section but also legends should systematically mention the nature of mutations (loss/gain), in the current version it is sometimes clarified but not consistently.

【Response】 Thank you very much for your advice. We have provided detailed explanations of the nature of mutations (loss/gain) in the figure captions, such as in Fig. 1g, Fig. 2a, and Fig 4a.

Legends should be improved and all need to be revised to have every acronym mentioned in the figure in full. In addition, the concentration of ethylene, as well as of 1-MCP, has to be mentioned in every figure legend, as well as in the method description.

【Response】 Thank you very much for your advice. We have used full terms as much as possible in the figures, such as “ethylene” instead of “ET”. When abbreviations are used, we have provided detailed explanations in the figure captions, for example, λ -PPase: Lambda Protein Phosphatase and 1-MCP: 1-Methylcyclopropene, and so forth.

We have annotated the concentrations and treatment times of ethylene and 1-MCP used in the figure captions. Furthermore, in the experimental methods section (Lines 542-544), we have also provided details regarding the concentrations and durations of ethylene and 1-MCP used.

Legend to Figure 3: Y2H for membrane based proteins: legend lacks information for the reader not familiar with the split ubiquitin systems. The authors also failed to refer to the original paper by Stagljar in PNAS in the manuscript, which has to be included. Again, all acronyms used in the figure panels are to be included in full in the legend. LCI assays are used in several experiments; mention Luciferase complementation assays in full in legends. Still on the legend to Figure 3. For non-specialist readers it is also more informative to indicate the proper domain structure of ethylene receptors with Ethylene binding -GAF- His-kinase – Receiver domain. The legend does not mention what REC is, and His-KA is not a standard acronym for Histidine kinase, rather HK. Clarify in legend and again, acronyms are to be mentioned in full.

【Response】 Thank you for your suggestion. We have provided a more detailed explanation of the terms related to the split-ubiquitin membrane yeast two-hybrid System (MbYTH), Cub, NubI, and NubG, in the caption of Fig. 4b. Additionally, we have cited the relevant reference by Stagljar et al. in line 230 of the manuscript.

We have expanded all abbreviations to their full terms in all figure legends. We have also labeled the full names of the structural domains of the ethylene receptors (Fig. 3e). Additionally, in the figure captions, we have displayed the full names of the abbreviations.

Minor:

1. Line 35: the ethylene receptor/CTR1 complex

【Response】 Thank you very much for your advice. The “CTR” here includes a wider range of plant CTR proteins in addition to the extensive Arabidopsis CTR1 studied, and we have modified the CTR here to CTRs.

2. Line 105: the sentence should be moved up

【Response】 Thank you very much for your advice. We have rearranged the order of the preceding and subsequent sentences (Lines 104-108).

3. Line 360: refers to AtCTR1, rather than using CTR1 without reference to the species. The latter can be very confusing since OsCTR1 exists too, yet OsCTR2 is the closest homolog of AtCTR1, not OsCTR1. A general check of the manuscript on this should be performed.

【Response】 Thank you very much for your advice. We have revised all references to Arabidopsis CTR1 in this manuscript to AtCTR1 for better differentiation from rice OsCTR1/2.

4. Line 416: this statement linking the predominant role of ETR2 in regulating OsCTR2 phosphorylation to the presence of a receiver domain, seems to lack a reference, since this manuscript does not present evidence for the binding of ETR1 neither EIN4.

【Response】 Thank you for your reminder. We have corrected the citations of the relevant references (Line 438). In the article by Clark et al.¹, it is explained that the binding of AtETR1, containing the receiver domain, with AtCTR1 is stronger than that

of AtETR1 and AtERS without the receiver domain.

5. Supplemental figures 12 and 13 contain information that deserves to be part of the core manuscript. Authors could combine S12 for instance with Fig1. Fig.S13 could be part of Fig.6.

【Response】 Thank you very much for your advice. We have relocated the results from the original Supplementary Fig. 13 to the core conclusion position in Fig. 6g. Because the original Supplementary Fig. 12 only provides transcriptional information, it remains available as Supplementary Fig. 15. We hope the reviewer would agree with us at this point.

6. Figure 3: ‘Prey’ is with an ‘e’ in this context, not ‘a.’

【Response】 Thank you so much for your thorough review of our manuscript. We have corrected this error in Fig. 3b.

7. The origin of all antibodies used in immunoblots should be clearly mentioned in the Materials section, including company or reference from lab of origin.

【Response】 Thank you for your advice. We have provided detailed explanations of the preparation method for antibodies, the relevant product numbers for commercial antibodies, and the concentrations used in our experiments in the Materials and Methods section (Lines 566-597).

Thank you very much for your valuable comments and other relevant parts were also revised accordingly.

For Reviewer #2 (Remarks to the Author):

In this paper, the authors reveal that MHZ3 is indispensable for the ethylene receptor-

mediated phosphorylation of OsCTR2 in rice, which seems important for ethylene signaling initiation. The authors found that MHZ3 could interact with ethylene receptors and facilitate the interactions between receptors and OsCTR2, which were inhibited by ethylene treatment. Uncovering the mechanistic role of MHZ3 in ethylene signaling is valuable since *mhz3* mutants are completely insensitive to ethylene both in coleoptile and root. Overall, the results presented in this manuscript are well-organized and convincing. However, there are some key questions mainly regard to the biological significance of this regulation remained to be addressed.

【Response】 Thank you very much for the valuable comments and we have addressed the issues in the following parts.

Major concerns:

1. Zhao et al., (Plant cell, 2020) had already reported the existence of OsCTR2 phosphorylation and ethylene treatment decreased the phosphorylation level of OsCTR2. In this paper, the author further confirmed the rapid phosphorylation changes of OsCTR2 upon the switch on/off of ethylene signaling cascade. It's very likely that the phosphorylation status of OsCTR2 may affect its kinase activity and thus regulate the downstream ethylene signaling. However, the authors only demonstrated a correlation between OsCTR2 phosphorylation and ethylene response, there are no direct results supporting the importance of OsCTR2 phosphorylation. The authors should identify the phosphorylation sites of OsCTR2 and construct related genetic materials to address the biological relevance of OsCTR2 phosphorylation in ethylene signaling.

【Response】 Thank you very much for your advice. Mayerhofer et al. (2012)² identified the autophosphorylation sites of Arabidopsis AtCTR1 (S703/T704/S707/S710) using mass spectrometry and structural biology methods. AtCTR1 functions as a dimer. Two inactive mutants of the activation loop, harboring the T704A/S707A and T704A/S707A/S710A, predominantly form monomers. Park et al.(2023)³ further demonstrated that mutations in the autophosphorylation sites (T704/S707/S710) of AtCTR1 prevent dimer formation and abolish its ability to phosphorylate EIN2. We aligned the amino acid sequences of OsCTR2 with that of

AtCTR1, identifying the autophosphorylation sites of OsCTR2 (T665/S668/S671) (Supplementary Fig. 2a). We introduced mutations in OsCTR2 at the corresponding residues, replacing them with Ala to generate the phosphor-dead mutant (OsCTR2^{AAA}). In both rice WT and *Osctr2* protoplasts, OsCTR2^{D-E} (kinase-dead) and OsCTR2^{AAA} (phosphor-dead) showed no detectable phosphorylation compared to wild-type OsCTR2 and lost their inhibitory effect on ethylene-responsive genes (Supplementary Figs. 2b, c). Additionally, we observed the absence of phosphorylation of OsCTR2 in the *Osetr2 ers2* double mutant (Fig. 2a), and mutations in both rice OsETR2 and OsERS2 result in ethylene hypersensitivity⁴, implying a connection between OsCTR2 autophosphorylation and ethylene hypersensitive phenotypes. In summary, these results indicate that the autophosphorylation of OsCTR2 is important for its function (Lines 133-141).

2. It seems that the non-phosphorylated OsCTR2 is inactive and could release the repression of downstream ethylene signaling through OsEIN2. The authors found that OsCTR2 phosphorylation is completely abolished by *mhz3* mutation in the air and couldn't respond to ethylene or 1-MCP treatment as well. Theoretically, ethylene signaling should be constitutively activated in *mhz3* mutants. However, the ethylene response is deprived in *mhz3*.

The authors claimed that mutation of MHZ3 could also result in the ubiquitination-mediated degradation of OsEIN2 as reported by Ma et al., 2018, which is epistatic to OsCTR2. It's reasonable, however, weaken the significance of the regulation of MHZ3 on OsCTR2. The authors provide strong evidences demonstrating that MHZ3 is indispensable for OsCTR2 phosphorylation, but, likewise, the biological relevance of this regulation is unclear. The authors should try to distinguish the two opposite roles of MHZ3 genetically.

【Response】 Thank you very much for your comments. Regarding the effects of OsCTR2 phosphorylation loss in the *mhz3* mutants, it should have some constitutive ethylene response phenotype somewhere theoretically. It is interesting to find that, compared to the WT, the length of coleoptiles in *mhz3-1* treated with the ethylene perception inhibitor 1-MCP for three days was not inhibited but significantly longer than the wild type with the same treatment (Supplementary Fig. 4a). This phenomenon

most likely reflects the residual constitutive ethylene response phenotype after losing the OsCTR2 phosphorylation activity in *mhz3*. These have been incorporated into the discussion part at lines 480-484.

Additionally, Overexpressing OsEIN2-GFP and OsEIL1-GFP in *mhz3* mutant resulted in a constitutively ethylene-responsive phenotype; however, these transgenic lines no longer respond to exogenous ethylene or 1-MCP treatment (Supplementary Fig. 16a, b), suggesting that the ethylene receptors have completely lost their ability to sense ethylene in *mhz3* mutant (Lines 485-490). This could be the role of MHZ3 in influencing ethylene receptor function.

Minor concerns:

1. Figure 3a, the resolution is not enough to distinctly show the colocalization of OsMHZ3 and ethylene receptors. The authors should provide higher resolution images.

【Response】 Thank you very much for your advice. We have provided higher-resolution images (Fig. 3a), and furthermore, we have replicated this experiment in rice protoplasts (Supplementary Fig. 9).

2. Figure 3c and figure 5d, there are additional GFP bands in the experimental materials expressed MHZ3-GFP, sometimes in input, sometimes in the IP product, why?

【Response】 Thank you for the comment. We noted that proteins fused with GFP tags sometimes break at the GFP tag during protein denaturation, and whether GFP can be detected depends on the abundance of GFP-fused proteins, the higher the abundance, the easier to detect the broken-free GFP. Uncropped blots have been provided in the Source Data file.

3. Figure 3f, the authors should indicate which band represents the OsERS2-GAFHK in the input group.

【Response】 Thank you for your suggestion. We have annotated OsERS2-GAFHK (Fig. 3f) and provided an explanation in the figure caption.

4. Figure 4b upper panel, the quality is too low to distinguish the phosphorylated and non-phosphorylated CTR1 especially in WT. The authors should replace this figure with better quality images.

【Response】 Thank you for your suggestion. We have replaced the original images with high-quality ones (Fig. 4b).

5. Figure 4c and d, the phosphorylation status of CTR1 in WT should be added to indicate the promotion effects of ethylene receptors overexpression.

【Response】 Thank you for your very helpful suggestions. We have included the wild-type control to demonstrate the promoting effect of ethylene receptor overexpression on OsCTR2 phosphorylation (Figs. 4c, d).

6. Line 171, identity. (Fig. 1i), “.” should be deleted.

【Response】 Thank you for the comment and we have corrected the error.

7. Line 190, “Under” should be “under”.

【Response】 Thank you for the comment and we have corrected this error in the revised manuscript (Line 202).

Thank you very much for your valuable comments and these made us understand the MHZ3 mechanisms more thoroughly than before. Other relevant parts were also revised accordingly.

Reference

1. Clark KL, Larsen PB, Wang X, Chang C. Association of the Arabidopsis CTR1 Raf-like kinase with the ETR1 and ERS ethylene receptors. *Proc Natl Acad Sci U S A* **95**, 5401-5406 (1998).
2. Mayerhofer H, Panneerselvam S, Mueller-Dieckmann J. Protein kinase domain of CTR1 from

Arabidopsis thaliana promotes ethylene receptor cross talk. *J Mol Biol* **415**, 768-779 (2012).

3. Park HL, *et al.* Ethylene-triggered subcellular trafficking of CTR1 enhances the response to ethylene gas. *Nature Communications* **14**, (2023).
4. Ma B, *et al.* Membrane protein MHZ3 stabilizes OsEIN2 in rice by interacting with its Nramp-like domain. *Proc Natl Acad Sci U S A* **115**, 2520-2525 (2018).

Reviewers' Comments:

Reviewer #2:

Remarks to the Author:

In this revised manuscript, the authors have resolved most of my previous concerns. They have examined the importance of OsCTR2 autophosphorylation in ethylene response and also collected some genetic evidences supporting the biological significance of MHZ3-regulated phosphorylation of OsCTR2. Overall, this is a well implemented study and could advance our understanding of the early events in ethylene signal transduction. I still have some minor concerns mainly regarding to the discussion part.

1.Line 489-493, overexpression of OsEIN2 or OsEIL1 in *mhz3* are insensitive to ethylene and MCP. The former could indicate that MHZ3 may affect the function of ethylene receptors; however, the insensitivity of OsEIL1ox/*mhz3* is not necessarily suggest an effect on the receptor. Based on the published study (Ma et al., PNAS, 2018), OsEIL1ox/*mhz3* may resemble OsEIL1ox/*ein2* due to the extremely low protein level of EIN2, which may cause the insensitivity to ethylene or MCP. Therefore, the phenotypes of OsEIL1ox/*mhz3* could not indicate the regulation of MHZ3 on ethylene receptors.

2.Line 497-502, the related discussion is inaccurate. The mild phenotype of MHZ3ox is not indeed caused by OsEIN2 protein degradation due to the enhanced activity of OsCTR2. The results presented in Ma et al., 2018 clearly showed that the protein level of OsEIN2 in MHZ3ox is much higher than WT. Although phosphorylation might promote EIN2 protein degradation, there is currently no direct evidence to confirm this definitively. On the contrary, it is generally believed that EIN2 phosphorylation prevents its cleavage, causing it to be retained in the endoplasmic reticulum membrane and rendering it inactive. Based on the results in this manuscript, MHZ3 may inhibit OsEIN2 activity through enhancing OsCTR2-mediated phosphorylation of OsEIN2. The antagonistic regulation of MHZ3 on OsEIN2 protein stability and activity may indeed result in the mild phenotype of MHZ3ox. This also provides another piece of genetic evidence to support the regulation of MHZ3 on OsCTR2 phosphorylation.

3.About the Figure 7, in the air (left panel), it is more accurate to replace "degradation" with "inactive". As mentioned above, the relationship between EIN2 phosphorylation and degradation remains unclear so far. More importantly, this "phosphorylation to degradation" model contradicts the previous research from the authors' group. If the OsCTR2-mediated OsEIN2 phosphorylation is crucial for its degradation, OsEIN2 should be more stable in *mhz3* since OsCTR2 was kept in the non-phosphorylated inactive form. According to the current knowledge, the phosphorylation of OsEIN2 also should be largely abolished in *mhz3*. However, OsEIN2 protein is very unstable and undergoes the proteasome-mediated degradation in *mhz3*. Hence, whether phosphorylation could promote EIN2 degradation still need more evidences.

4.The regulation of MHZ3 on receptors' function is quite interesting. Since MHZ3 only interacts with the TM and GAF domains of receptors, it's very likely that MHZ3 may have a direct impact on receptor function either the conformation or activity. Whether MHZ3 could directly affect the binding ability of receptors to ethylene and MCP is worth to be addressed. In this case, MHZ3 may work similarly with RAN1 in arabidopsis, which is crucial for the biogenesis and binding activity of ethylene receptors through regulating copper delivery. *ran1ein2* double mutant showed similar phenotype with *ein2* in the dark, while exhibited severe growth defects resembling *ran1* single mutant at latter growth stages under light (Woeste et al., Plant cell, 2000). Therefore, it is also very interesting to examine the phenotypes of *mhz3* at different growth stages to get a comprehensive understanding of MHZ3 function beyond ethylene signaling. The authors should add discussions about the possible mechanisms underlying the MHZ3's impact on ethylene receptors.

5.Figure 3a, the resolution is still too low to show the reticulum structure of the ER, especially the

confocal image of MHZ3-GFP. The authors should refer to the images presented in Ma et al., 2018 which clearly showed the ER-localization of MHZ3-GFP, and provide higher-quality images.

Responses to the REVIEWERS COMMENTS

Reviewer #2 (Remarks to the Author):

In this revised manuscript, the authors have resolved most of my previous concerns. They have examined the importance of OsCTR2 autophosphorylation in ethylene response and also collected some genetic evidences supporting the biological significance of MHZ3-regulated phosphorylation of OsCTR2. Overall, this is a well implemented study and could advance our understanding of the early events in ethylene signal transduction. I still have some minor concerns mainly regarding to the discussion part.

【Response】 Thank you very much for your valuable comments. We have addressed the issues in the following sections.

1.Line 489-493, overexpression of OsEIN2 or OsEIL1 in mhz3 are insensitive to ethylene and MCP. The former could indicate that MHZ3 may affect the function of ethylene receptors; however, the insensitivity of OsEIL1ox/mhz3 is not necessarily suggest an effect on the receptor. Based on the published study (Ma et al., PNAS, 2018), OsEIL1ox/mhz3 may resemble OsEIL1ox/ein2 due to the extremely low protein level of EIN2, which may cause the insensitivity to ethylene or MCP. Therefore, the phenotypes of OsEIL1ox/mhz3 could not indicate the regulation of MHZ3 on ethylene receptors.

【Response】 Thank you for your very helpful suggestions. We have made changes in Supplementary Fig. 17 to more accurately highlight the impact of MHZ3 on ethylene receptor function. Additionally, we have revised lines 491-496 in the discussion section.

2.Line 497-502, the related discussion is inaccurate. The mild phenotype of MHZ3ox is not indeed caused by OsEIN2 protein degradation due to the enhanced activity of OsCTR2. The results presented in Ma et al., 2018 clearly showed that the protein level of OsEIN2 in MHZ3ox is much higher than WT. Although phosphorylation might promote EIN2 protein degradation, there is currently no direct evidence to confirm this

definitively. On the contrary, it is generally believed that EIN2 phosphorylation prevents its cleavage, causing it to be retained in the endoplasmic reticulum membrane and rendering it inactive. Based on the results in this manuscript, MHZ3 may inhibit OsEIN2 activity through enhancing OsCTR2-mediated phosphorylation of OsEIN2. The antagonistic regulation of MHZ3 on OsEIN2 protein stability and activity may indeed result in the mild phenotype of MHZ3ox. This also provides another piece of genetic evidence to support the regulation of MHZ3 on OsCTR2 phosphorylation.

【Response】 Thank you for your thorough review of our manuscript. We have made revisions to lines 498-505 in the discussion section based on your suggestions, which have made our statements more precise.

3. About the Figure 7, in the air (left panel), it is more accurate to replace “degradation” with “inactive”. As mentioned above, the relationship between EIN2 phosphorylation and degradation remains unclear so far. More importantly, this “phosphorylation to degradation” model contradicts the previous research from the authors’ group. If the OsCTR2-mediated OsEIN2 phosphorylation is crucial for its degradation, OsEIN2 should be more stable in mhz3 since OsCTR2 was kept in the non-phosphorylated inactive form. According to the current knowledge, the phosphorylation of OsEIN2 also should be largely abolished in mhz3. However, OsEIN2 protein is very unstable and undergoes the proteasome-mediated degradation in mhz3. Hence, whether phosphorylation could promote EIN2 degradation still need more evidences.

【Response】 Thank you for your suggestions. We have made changes in Fig. 7 and updated the corresponding legend. Additionally, we have discussed the relevant content in lines 507-515.

4. The regulation of MHZ3 on receptors’ function is quite interesting. Since MHZ3 only interacts with the TM and GAF domains of receptors, it’s very likely that MHZ3 may have a direct impact on receptor function either the conformation or activity. Whether MHZ3 could directly affect the binding ability of receptors to ethylene and MCP is worth to be addressed. In this case, MHZ3 may work similarly with RAN1 in arabidopsis, which is crucial for the biogenesis and binding activity of ethylene

receptors through regulating copper delivery. *ran1ein2* double mutant showed similar phenotype with *ein2* in the dark, while exhibited severe growth defects resembling *ran1* single mutant at latter growth stages under light (Woeste et al., Plant cell, 2000). Therefore, it is also very interesting to examine the phenotypes of *mhz3* at different growth stages to get a comprehensive understanding of MHZ3 function beyond ethylene signaling. The authors should add discussions about the possible mechanisms underlying the MHZ3's impact on ethylene receptors.

【Response】 Thank you for your comments and suggestions. Regarding the regulation of ethylene receptor function by MHZ3, we have added relevant discussion in lines 461-465.

5. Figure 3a, the resolution is still too low to show the reticulum structure of the ER, especially the confocal image of MHZ3-GFP. The authors should refer to the images presented in Ma et al., 2018 which clearly showed the ER-localization of MHZ3-GFP, and provide higher-quality images.

【Response】 Thank you very much for your suggestion. We have provided higher resolution images to more clearly demonstrate the colocalization of the proteins (Fig. 3a).

Thank you very much for your valuable comments and these made us understand the MHZ3 mechanisms more thoroughly than before. Other relevant parts were also revised accordingly.